

# Model and proxy evidence for coordinated changes in the hydroclimate of distant regions over the Last Millennium

Pedro José Roldán-Gómez[1], Jesús Fidel González-Rouco[1], Jason E. Smerdon[2], and Félix García-Pereira[1]

[1]Instituto de Geociencias, Consejo Superior de Investigaciones Científicas - Universidad Complutense de Madrid, 28040 Madrid, Spain

[2]Lamont-Doherty Earth Observatory of Columbia University, Palisades, NY, United States of America

**Correspondence:** P. J. Roldán-Gómez (peroldan@ucm.es)

**Abstract.** The Medieval Climate Anomaly (MCA; ca. 950-1250 CE) and the Little Ice Age (LIA; ca. 1450-1850 CE) periods, generally characterised by respectively higher and lower temperatures in many regions, have also been associated with drier and wetter conditions in areas around the Intertropical Convergence Zone (ITCZ), the Asian Monsoon region, and in areas impacted by large-scale climatic modes like the Northern and Southern Annular Modes (NAM and SAM, respectively). To

analyse coordinated changes in large-scale hydroclimate patterns, and whether similar changes also extend to other periods of the Last Millennium (LM) outside the MCA and the LIA, reconstruction-based products have been analysed, including the collection of tree-ring based Drought Atlases (DA), the Paleo Hydrodynamics Data Assimilation product (PHYDA) and the Last Millennium Reanalysis (LMR). These analyses have shown coherent changes in the hydroclimate of tropical and extratropical regions, such as northern and central South America, East Africa, western North America, Western Europe, the

Middle East, Southeast Asia and the Indo-Pacific, during the MCA, the LIA and other periods of the LM. Comparisons with model simulations from the Community Earth System Model - Last Millennium Ensemble (CESM-LME) and phases 5 and 6 of the Coupled Model Intercomparison Project (CMIP5 and CMIP6) show that both external forcing and internal variability contributed to these changes, with the contribution of internal variability being particularly important in the Indo-Pacific basin and that of external forcing in the Atlantic basin. These results may help to identify not only those areas showing coordinated

changes, but also those regions where model simulations are able to successfully reproduce the evolution of hydroclimate during the LM.

## 1 Introduction

The climate of the Last Millennium (LM) was characterised in many regions by periods of warmer and cooler conditions with respect to the pre-industrial mean climate (Diaz et al., 2011; Graham et al., 2010), two well-known examples being the

Medieval Climate Anomaly (MCA; ca. 950-1250 CE) and the Little Ice Age (LIA; ca. 1450-1850 CE; Graham et al., 2007; Laird et al., 2012; Ledru et al., 2013). Such temperature changes are often associated with changes in external forcing factors like solar variability and volcanic activity (Mann et al., 2009; Schurer et al., 2013; Fernández-Donado et al., 2013). The impact of these changes on hydroclimate is not straightforward, but reconstructions show that episodes of severe drought also took place during the LM in many regions (Cook et al., 2022). Periods of wetter and drier conditions existed in large areas of North





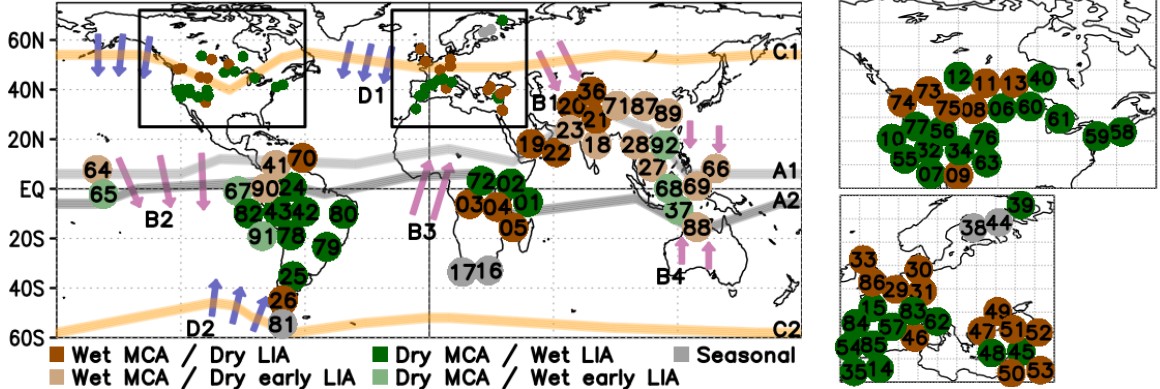

**Figure 1.** Hydroclimate reconstructions showing **(dark brown)** drier conditions during MCA and wetter conditions during LIA, **(light brown)** drier conditions during MCA and wetter conditions during early LIA (until late 1500s CE), **(dark green)** wetter conditions during MCA and drier conditions during LIA, **(light green)** wetter conditions during MCA and drier conditions during early LIA (until late 1500s CE), and **(gray)** drier or wetter conditions in MCA and LIA depending on the season. Detailed references are included in Table 1. As per references of Table 2, A1 and A2 respectively indicate the current position of the ITCZ in July and January, B1 to B4 the ITCZ changes in the transition from MCA to LIA, C1 and C2 the boundary between low and high pressures in the pattern of NAM and SAM, and D1 and D2 the changes in this boundary in the transition from MCA to LIA. Regional maps over North America and Europe are shown on the right side of the panel.

America (Cook et al., 2010b), Europe (Luterbacher et al., 2012), South America (Vuille et al., 2012) and Monsoon Asia (Hu et al., 2008), with occurrences that coincide with the MCA and the LIA.

To survey the available evidence, we assessed 92 reconstructions reporting changes from wetter to drier or from drier to wetter conditions during the transition from the MCA to LIA (Fig. 1, with detailed references in Table 1). Some of these reconstructions show consistent drier or wetter conditions during the whole LIA (dark green and dark brown), while others show drier or wetter conditions until late 1500s CE, during the early LIA (light green and light brown), as reported for areas of Pakistan, western India and southern China by Graham et al. (2010). These similarities in the evolution of the hydroclimate of distant regions suggest coordinated changes at a global scale (Graham et al., 2010; Atwood et al., 2021). For example, reconstructions from southwestern North America (Graham et al., 2007; Meko et al., 2001; Cook et al., 2004, 2010b; Hughes and Funkhouser, 1998; Anderson, 2011) and the Mediterranean basin (Luterbacher et al., 2012; Martín-Puertas et al., 2010; Morellón et al., 2009) show drier conditions during the MCA and wetter conditions during the LIA, while reconstructions from northwestern North America (Steinman et al., 2013; Stevens and Dean, 2008), central Europe (Büntgen et al., 2010, 2011) and the British Isles (Wilson et al., 2012; Proctor et al., 2000) tend to show wetter conditions during the MCA and drier conditions during the LIA.

An antiphased relationship between the MCA and LIA is also observed in tropical areas of South America, with wetter conditions during the MCA in areas of northern South America, like the Cariaco Basin in Venezuela (Haug et al., 2001),



**Table 1.** Location, references and type of reconstruction (sediments, tree rings, speleothems, documentary, ice cores or multi-proxy) for the reconstructions represented in Fig. 1. The first column indicates the code used in Fig. 1 for each reconstruction.

| Code | Location | References | Type |
|---|---|---|---|
| 01 | Lake Challa (Tanzania) | Verschuren et al. (2009); Wolff et al. (2011) | Sediments |
| 02 | Lake Naivasha (Kenya) | Verschuren et al. (2000); Verschuren (2001) | Sediments |
| 03 | Lake Tanganyika (Tanzania, DRC) | Tierney et al. (2010a) | Sediments |
| 04 | Lake Masoko (Tanzania) | Gilbert et al. (2002); Garcin et al. (2006, 2007) | Sediments |
| 05 | Lake Malawi (Malawi) | Johnson et al. (2001); Brown and Johnson (2005); Johnson and McCave (2008) | Sediments |
| 06 | Moon Lake (North Dakota, USA) | Laird et al. (1996) | Sediments |
| 07 | California (USA) | Graham et al. (2007) | Multi-proxy |
| 08 | Bighorn (Wyoming, USA) | Gray et al. (2004) | Tree rings |
| 09 | Malpais (New Mexico, USA) | Grissino-Mayer (1995) | Tree rings |
| 10 | Sacramento River (California, USA) | Meko et al. (2001) | Tree rings |
| 11 | Humboldt Lake (Canada) | Michels et al. (2007) | Sediments |
| 12 | North Saskatchewan River (Canada) | Case and MacDonald (2003) | Tree rings |
| 13 | Red River (Canada) | St.George and Nielsen (2002) | Tree rings |
| 14 | Guadalentín River (Spain) | Benito et al. (2010) | Sediments |
| 15 | Gardon River (France) | Sheffer et al. (2007) | Sediments |
| 16 | Orange River (South Africa) | Tyson and Lindesay (1992) | Sediments |
| 17 | Buffels River (W South Africa) | Benito et al. (2011) | Sediments |
| 18 | Dandak Cave (India) | Sinha et al. (2007); Berkelhammer et al. (2010) | Speleothems |
| 19 | S Oman | Burns et al. (2002) | Speleothems |
| 20 | Lunkaransar Lake (India) | Bryson and Swain (1981) | Sediments |
| 21 | Pindar Valley (India) | Phadtare and Pant (2006) | Sediments |
| 22 | Arabian Sea (Oman) | Anderson et al. (2002); Gupta et al. (2003) | Sediments |
| 23 | Karachi (Pakistan) | von Rad et al. (1999) | Sediments |
| 24 | Lake Pumacocha (Peru) | Bird et al. (2011) | Sediments |
| 25 | N Argentina | Boucher et al. (2011) | Multi-proxy |
| 26 | Patagonia | Boucher et al. (2011) | Multi-proxy |
| 27 | S Vietnam | Buckley et al. (2010) | Tree rings |
| 28 | NW Thailand | Buckley et al. (2007) | Tree rings |
| 29 | NE France | Büntgen et al. (2011) | Tree rings |
| 30 | N Germany | Büntgen et al. (2010, 2011) | Tree rings |





| Code | Location | References | Type |
|---|---|---|---|
| 31 | S Germany | Büntgen et al. (2010, 2011) | Tree rings |
| 32 | SW North America | Cook et al. (2004, 2010b) | Tree rings |
| 33 | Uamh Cave (NW Scotland) | Proctor et al. (2000) | Speleothems |
| 34 | Arizona and Utah (USA) | Ely et al. (1993) | Sediments |
| 35 | Morocco | Esper et al. (2007) | Tree rings |
| 36 | Karakorum Mountains (Pakistan) | Treydte et al. (2006) | Tree rings |
| 37 | Liang Luar Cave (Indonesia) | Griffiths et al. (2016) | Speleothems |
| 38 | Finland | Helama et al. (2009) | Tree rings |
| 39 | Kola Peninsula (Russia) | Kremenetski et al. (2004) | Multi-proxy |
| 40 | NW Ontario (Canada) | Laird et al. (2012) | Sediments |
| 41 | Papallacta (Ecuador) | Ledru et al. (2013) | Sediments |
| 42 | Marcacocha (S Peru) | Chepstow-Lusty et al. (2009) | Sediments |
| 43 | Cascayunga (Peru) | Reuter et al. (2009) | Speleothems |
| 44 | Saavanjoki River (Finland) | Luoto et al. (2013) | Sediments |
| 45 | Antalya and Mersin (SW Turkey) | Touchan et al. (2007) | Tree rings |
| 46 | Grotta Verde (Italy) | Antonioli et al. (2003) | Speleothems |
| 47 | Urzuntala Cave (NW Turkey) | Göktürk (2011) | Speleothems |
| 48 | Kocain Cave (S Turkey) | Göktürk (2011) | Speleothems |
| 49 | Sofular Cave (N Turkey) | Göktürk et al. (2011) | Speleothems |
| 50 | Nahal Zin (Israel) | Greenbaum et al. (2000) | Sediments |
| 51 | Nar Gölü (C Turkey) | Jones et al. (2006); Woodbridge and Roberts (2011) | Sediments |
| 52 | Tercer Lake (C Turkey) | Kuzucuoglu et al. (2011) | Sediments |
| 53 | Dead Sea | Bookman et al. (2004); Neumann et al. (2007) | Sediments |
| 54 | Zoñar Lake (S Spain) | Martín-Puertas et al. (2008, 2010) | Sediments |
| 55 | Sierra Nevada (California, USA) | Stine (1994) | Tree rings |
| 56 | Great Basin (W North America) | Hughes and Funkhouser (1998) | Tree rings |
| 57 | NE Spain | Llasat et al. (2003) | Documentary |
| 58 | Little Pond (Massachusetts, USA) | Oswald and Foster (2011) | Sediments |
| 59 | Piermont Marsh (New York, USA) | Pederson et al. (2005) | Sediments |
| 60 | Hole Bog (Minnesota, USA) | Booth et al. (2006) | Sediments |
| 61 | Minden Bog (Michigan, USA) | Booth et al. (2006) | Sediments |
| 62 | Lake Accesa (Italy) | Magny et al. (2007) | Sediments |
| 63 | San Juan Mountains (Colorado, USA) | Routson et al. (2011) | Tree rings |
| 64 | Teraina (Kiribati) | Sachs et al. (2009) | Sediments |



| Code | Location | References | Type |
|------|----------|-----------|------|
| 65 | Kiritimati (Kiribati) | Sachs et al. (2009); Higley et al. (2018) | Sediments |
| 66 | Mecherchar Island (Palau) | Sachs et al. (2009) | Sediments |
| 67 | San Cristóbal Island (Ecuador) | Sachs et al. (2009) | Sediments |
| 68 | Makassar Strait (Indonesia) | Newton et al. (2006); Tierney et al. (2010b) | Sediments |
| 69 | Kau Bay (Indonesia) | Langton et al. (2008) | Sediments |
| 70 | Cariaco Basin (Venezuela) | Haug et al. (2001) | Sediments |
| 71 | Dongge Cave (S China) | Wang et al. (2005) | Speleothems |
| 72 | Lake Victoria (Tanzania, Uganda, Kenya) | Stager et al. (2005) | Sediments |
| 73 | Castor Lake (Washington, USA) | Steinman et al. (2013) | Sediments |
| 74 | Lime Lake (Washington, USA) | Steinman et al. (2013) | Sediments |
| 75 | Crevice Lake (Montana, USA) | Stevens and Dean (2008) | Sediments |
| 76 | Bison Lake (Colorado, USA) | Anderson (2011) | Sediments |
| 77 | Pyramid Lake (Nevada, USA) | Benson et al. (2002) | Sediments |
| 78 | Quelccaya Ice Cap (Peru) | Vuille et al. (2012); Thompson et al. (1986) | Ice cores |
| 79 | Cristal Cave (SE Brazil) | Vuille et al. (2012); Taylor (2010) | Speleothems |
| 80 | Diva & Torrinha Caves (NE Brazil) | Vuille et al. (2012); Novello et al. (2012) | Speleothems |
| 81 | S South America | Neukom et al. (2010) | Multi-proxy |
| 82 | Palestina Cave (NW Peru) | Apaéstegui et al. (2014) | Speleothems |
| 83 | Lake Allos (France) | Wilhelm et al. (2012) | Sediments |
| 84 | Lake Taravilla (Spain) | Moreno et al. (2008) | Sediments |
| 85 | Lake Estanya (NE Spain) | Morellón et al. (2009, 2011) | Sediments |
| 86 | SC England | Wilson et al. (2012) | Tree rings |
| 87 | Wanxiang Cave (China) | Zhang et al. (2008) | Speleothems |
| 88 | Cave KNI-51 (N Australia) | Denniston et al. (2016) | Speleothems |
| 89 | Heshang Cave (China) | Hu et al. (2008) | Speleothems |
| 90 | Laguna Pallcacocha (S Ecuador) | Moy et al. (2002) | Sediments |
| 91 | Peruvian Shelf (Peru) | Rein et al. (2004) | Sediments |
| 92 | Dongdao Island (S China Sea) | Yan et al. (2011) | Sediments |

and drier conditions in areas of central South America, like Peru (Bird et al., 2011; Reuter et al., 2009; Vuille et al., 2012; Thompson et al., 1986; Apaéstegui et al., 2014) and Eastern Brazil (Vuille et al., 2012; Taylor, 2010; Novello et al., 2012). In East Africa, Anchukaitis and Tierney (2013) showed coordinated changes for lake Challa (Verschuren et al., 2009; Wolff et al., 2011), Naivasha (Verschuren et al., 2000; Verschuren, 2001), Tanganyika (Tierney et al., 2010a), Masoko (Gilbert et al., 2002; Garcin et al., 2006, 2007) and Malawi (Johnson et al., 2001; Brown and Johnson, 2005; Johnson and McCave, 2008), with an opposite behavior between the lakes in the south and the north of the Intertropical Convergence Zone (ITCZ). Changes in



**Table 2.** References for the position of ITCZ, the boundary between low and high pressures in the pattern of NAM and SAM, and the changes in these two elements during MCA and LIA, according to the codes included in Fig. 1.

| Code | Mode | References |
| --- | --- | --- |
| A1 | Modern position of July ITCZ | Newton et al. (2006) |
| A2 | Modern position of January ITCZ | Newton et al. (2006) |
| B1 | ITCZ changes in the Indian Monsoon region | Fleitmann et al. (2003) |
| B2 | ITCZ changes in the Eastern Pacific | Higley et al. (2018) |
| B3 | ITCZ changes in East Africa | Anchukaitis and Tierney (2013) |
| B4 | ITCZ changes in the Western Pacific | Denniston et al. (2016); Yan et al. (2015) |
| C1 | Boundary of low/high in the pattern of NAM | Li and Wang (2003) |
| C2 | Boundary of low/high in the pattern of SAM | Gong and Wang (1999) |
| D1 | Changes in NAM/NAO | Ortega et al. (2015) |
| D2 | Changes in SAM | Jones et al. (2009); Fogt et al. (2009) |

the ITCZ have been also associated with coordinated changes in the hydroclimate of the Indian Monsoon region, with wetter conditions during the MCA and drier conditions during the LIA for wide areas of Pakistan (von Rad et al., 1999; Treydte et al., 2006), India (Sinha et al., 2007; Berkelhammer et al., 2010; Bryson and Swain, 1981; Phadtare and Pant, 2006) and the

Arabian Sea (Burns et al., 2002; Fleitmann et al., 2003; Anderson et al., 2002; Gupta et al., 2003). Alterations of the ITCZ have been also associated with changes in the hydroclimate of the Indo-Pacific basin (Atwood et al., 2021), with a marked transition between the MCA and LIA in areas of China (Wang et al., 2005; Zhang et al., 2008; Hu et al., 2008), Southeast Asia (Buckley et al., 2010, 2007), Indonesia (Griffiths et al., 2016; Newton et al., 2006; Tierney et al., 2010b), northern Australia (Denniston et al., 2016) and the eastern Pacific islands (Sachs et al., 2009; Higley et al., 2018).

Most reconstructions in Fig. 1 are located in extratropical areas, in the boundary between low and high pressures within the regions of the Northern and Southern Annular Modes (NAM and SAM; Thompson and Wallace, 2001; Jones et al., 2009; Fogt et al., 2009), or in tropical areas around the ITCZ (Table 2). Analyses based on simulated data (Roldán-Gómez et al., 2020) link transitions from the MCA to LIA in extratropical areas with alterations in the variability of modes like the NAM and SAM (Ortega et al., 2015; Jones et al., 2009; Fogt et al., 2009), mainly driven by changes in external forcing factors (Roldán-

Gómez et al., 2020). In tropical areas, the transitions from the MCA to LIA can be linked to alterations of the ITCZ (Atwood et al., 2021), like a contraction over the Western Pacific (Denniston et al., 2016; Yan et al., 2015) and shifts over East Africa (Anchukaitis and Tierney, 2013), the Indian Monsoon region (Fleitmann et al., 2003) and the Eastern Pacific (Higley et al., 2018). The analysis of the simulated hydroclimate from the Community Earth System Model - Last Millennium Ensemble (CESM-LME; Otto-Bliesner et al., 2015) performed by Roldán-Gómez et al. (2022) shows that these alterations of the ITCZ

may be impacted by both external forcing factors and internal variability.





Even if individual reconstructions from different regions suggest coordinated changes in the hydroclimate during the MCA and LIA, global analyses based on proxy data are scarce and limited to certain reconstructions and products. The analysis of simulated data provides an insight into the mechanisms explaining coordinated changes in the hydroclimate of the LM, but analyses of LM simulations must account for both forced and internal variability (Roldán-Gómez et al., 2022). This work

addresses these challenges by performing a comparative assessment of the LM hydroclimate at global and large-continental scales from different sources, including reconstructions, model simulations, and hybrid products based on data assimilation and reanalysis. The use of several reconstruction-based products allows the identification with a robustness assessment of those regions more impacted by hydroclimatic changes during the MCA and LIA, and more generally during the LM. The comparison between reconstruction-based products and model simulations from different ensembles allows an assessment of

the contributions of external forcing and internal variability on the hydroclimate of each region, showing those regions where model simulations can provide representative information about changes in the hydroclimate.

## 2  Data and methods

The following reconstruction-based products are employed: the 100-member ensemble of the Paleo Hydrodynamics Data Assimilation product (PHYDA; Steiger et al., 2018), the 20-member ensemble of the Last Millennium Reanalysis (LMR;

Tardif et al., 2019) and the tree-ring based Drought Atlases (DA), including the Old World Drought Atlas (OWDA; Cook et al., 2015), the North American Drought Atlas (NADA; Cook et al., 2010b), the Monsoon Asia Drought Atlas (MADA; Cook et al., 2010a) and the Mexican Drought Atlas (MXDA; Stahle et al., 2016). Due to the fact that they do not extend back into the MCA, other DAs have not been included in our analyses.

Regarding the model-based products, we use the 13 all-forcing simulations of the CESM-LME, together with 10 LM sim-

ulations from the Coupled Model Intercomparison Project Phase 5 / Paleoclimate Modelling Intercomparison Project Phase 3 (CMIP5/PMIP3; Taylor et al., 2007; Schmidt et al., 2011, 2012; Stocker et al., 2013) and 4 LM simulations from the Coupled Model Intercomparison Project Phase 6 / Paleoclimate Modelling Intercomparison Project Phase 4 (CMIP6/PMIP4; Eyring et al., 2016; Jungclaus et al., 2017), all of which are described in Table 3. All these simulations have been interpolated to a common grid resolution of 6º×6º, the coarsest among the analysed simulations, to allow for the calculation of ensemble aver-

ages across the CESM-LME, CMIP5 and CMIP6 products. The CESM-LME and CMIP5 simulations are forced according to the CMIP5/PMIP3 protocol (Schmidt et al., 2011, 2012), while the CMIP6 simulations consider the CMIP6/PMIP4 protocol (Jungclaus et al., 2017).

Considering the wide bibliography of single-point reconstructions showing wetter or drier conditions during MCA and LIA (Fig. 1), most analyses have been based on these two periods. Following the approach of Masson-Delmotte et al. (2013), the

MCA has been considered from 950 to 1250 CE, and the LIA from 1450 to 1850 CE, even if these temporal intervals might not be suitable for all the regions and for all products (Neukom et al., 2014, 2019). To assess the robustness of our results to the specifics of this definition, we have performed additional analyses based on the trend during the whole period from 950 to 1750 CE, and report our findings in the Appendix A.



**Table 3.** Reconstruction-based and model products considered in this work. In case of being refered to an ensemble of experiments, the acronym used for the ensemble and the number of members (N) are provided. References describing each product are provided in the last column.

| Ensemble | Product/Model | N | References |
|---|---|---|---|
| DA | Old World Drought Atlas (OWDA) | 1 | Cook et al. (2015) |
| DA | North American Drought Atlas (NADA) | 1 | Cook et al. (2010b) |
| DA | Monsoon Asia Drought Atlas (MADA) | 1 | Cook et al. (2010a) |
| DA | Mexican Drought Atlas (MXDA) | 1 | Stahle et al. (2016) |
| LMR | Last Millennium Reanalysis (LMR) | 20 | Tardif et al. (2019) |
| PHYDA | Paleo Hydrodynamics Data Assimilation (PHYDA) | 100 | Steiger et al. (2018) |
| CESM-LME | Community Earth System Model (CESM) | 13 | Otto-Bliesner et al. (2015) |
| CMIP5 | Goddard Institute for Space Studies (GISS) | 3 | Schmidt et al. (2006, 2014) |
| CMIP5 | Model for Interdisciplinary Research on Climate (MIROC-ESM) | 1 | Watanabe et al. (2011) |
| CMIP5 | Hadley Centre Coupled Model (HadCM) | 1 | Tett et al. (2007) |
| CMIP5 | Community Climate System Model (CCSM) | 1 | Landrum et al. (2013) |
| CMIP5 | Institut Pierre Simon Laplace (IPSL) | 1 | Dufresne et al. (2013) |
| CMIP5 | Commonwealth Scientific and Industrial Research Organization (CSIRO) | 1 | Phipps et al. (2012) |
| CMIP5 | Meteorological Research Institute (MRI) | 1 | Yukimoto et al. (2011); Adachi et al. (2013) |
| CMIP5 | Max Planck Institut fur Meteorologie (MPI) | 1 | Giorgetta et al. (2013) |
| CMIP6 | MIROC Earth System version 2 for Long-term simulations (MIROC-ES2L) | 1 | Hajima et al. (2020) |
| CMIP6 | Meteorological Research Institute (MRI) | 1 | Yukimoto et al. (2019) |
| CMIP6 | Max Planck Institut fur Meteorologie (MPI) | 2 | Mauritsen et al. (2019) |





Our analyses focus on reconstructions of the Palmer Drought Severity Index (PDSI; Palmer, 1965). The PDSI is directly
provided by the DAs, the PHYDA and the LMR. For the CMIP5, CMIP6 and CESM-LME simulations, PDSI has been com-
puted based on atmospheric variables and soil parameters, following the approach of a self-calibrating PDSI (scPDSI; Wells
et al., 2004). The potential evapotranspiration is calculated using the Thornthwaite's method (Thornthwaite, 1948). While the
Thorntwaite formulation has been shown to have limitations for projections of 21st-century soil moisture conditions because
of its strong dependence on temperature, assessments of the Thornthwaite formulation in model simulations of the LM have
shown that such limitations are not a concern for the LM period (Smerdon et al., 2015).

The PHYDA provides water-year annual averages and seasonal averages for the boreal summer (June, July and August; JJA)
and winter (December, January, and February; DJF). The LMR only provides annual averages, while the Northern Hemisphere
DAs exclusively target the JJA period. For all the analyses, JJA values are considered for all the products except for the LMR,
for which the annual values are used. To confirm that the annual results from the LMR can be meaningfully compared to
the seasonal results from the other products, the annual PDSI from the PHYDA and the ensembles of CMIP5, CMIP6 and
CESM-LME are analysed in Appendix B.

Coordinated changes during the MCA and the LIA are characterized by computing the difference between each variable and
product during the LIA and MCA periods. The agreement between products is also assessed by computing the global spatial
correlation between PDSI LIA-MCA maps from each product (Coats et al., 2013a), and by counting the number of products
showing positive or negative PDSI differences for each location. Finally, time series of PDSI for the whole millennium are
extracted from different locations and different products, to analyse the correlations between them beyond the periods of the
MCA and LIA.

## 3 Results

### 3.1 Reconstructions of hydroclimate during the MCA and LIA

PDSI differences between the LIA and MCA are shown for the DAs, the PHYDA and the LMR (Fig. 2). Some wetting and
drying patterns are consistent across the DAs (Fig. 2a), the PHYDA (Fig. 2b), and the LMR (Fig. 2c). For example, all products
estimate drier conditions during the LIA and wetter conditions during the MCA over the Middle East. In the PHYDA and LMR
these conditions are associated with a warmer LIA in these regions (Fig. 3). There is also good agreement across the PHYDA
and LMR in South America, with drier conditions in the north and wetter conditions in the central continent during the LIA,
as well as in northern Canada, Scandinavia and northern Eurasia, with wetter conditions during the LIA, associated all of
them with lower temperatures (Fig. 3). The PHYDA and the LMR agree well with the reconstructions of Fig. 1, which also
show drier conditions during the LIA in southeastern Europe and Anatolia (Göktürk, 2011; Jones et al., 2006; Woodbridge and
Roberts, 2011; Kuzucuoglu et al., 2011), the Indian Monsoon region (Burns et al., 2002; Bryson and Swain, 1981; Phadtare
and Pant, 2006; Anderson et al., 2002; Gupta et al., 2003) and South America north of the ITCZ (Haug et al., 2001). The LMR
also agree with the reconstructions of Fig. 1 in regions of East Africa (Anchukaitis and Tierney, 2013).







**Figure 2.** LIA-MCA differences in PDSI (shading) or precipitation (contours) from **(a)** DAs, **(b)** PHYDA, and **(c)** LMR. Dots represent the locations of Fig. 1, with the same color code considered in that figure; positions of the modern ITCZ and NAM and SAM boundaries are also shown, as described in Fig. 1. For the precipitation, contours of -1 (brown) and 1 (green) mm/month are shown. Average of JJA is considered for the PHYDA and the DAs, while the annual average is considered for the LMR. For the PHYDA and the LMR, regional maps over North America and Europe are shown on the right side of each corresponding panel.

In western North America there is a good agreement between the DA and PHYDA, and between these two products and the reconstructions of Fig. 1, most of them showing wetter conditions in the southwest and drier conditions in the northwest during the LIA (Steinman et al., 2013; Cook et al., 2004, 2010b). Each of these products are nevertheless in contrast to the LMR, which shows drier conditions during the LIA for most of the American West, in sharp contrast with a broad range of literature





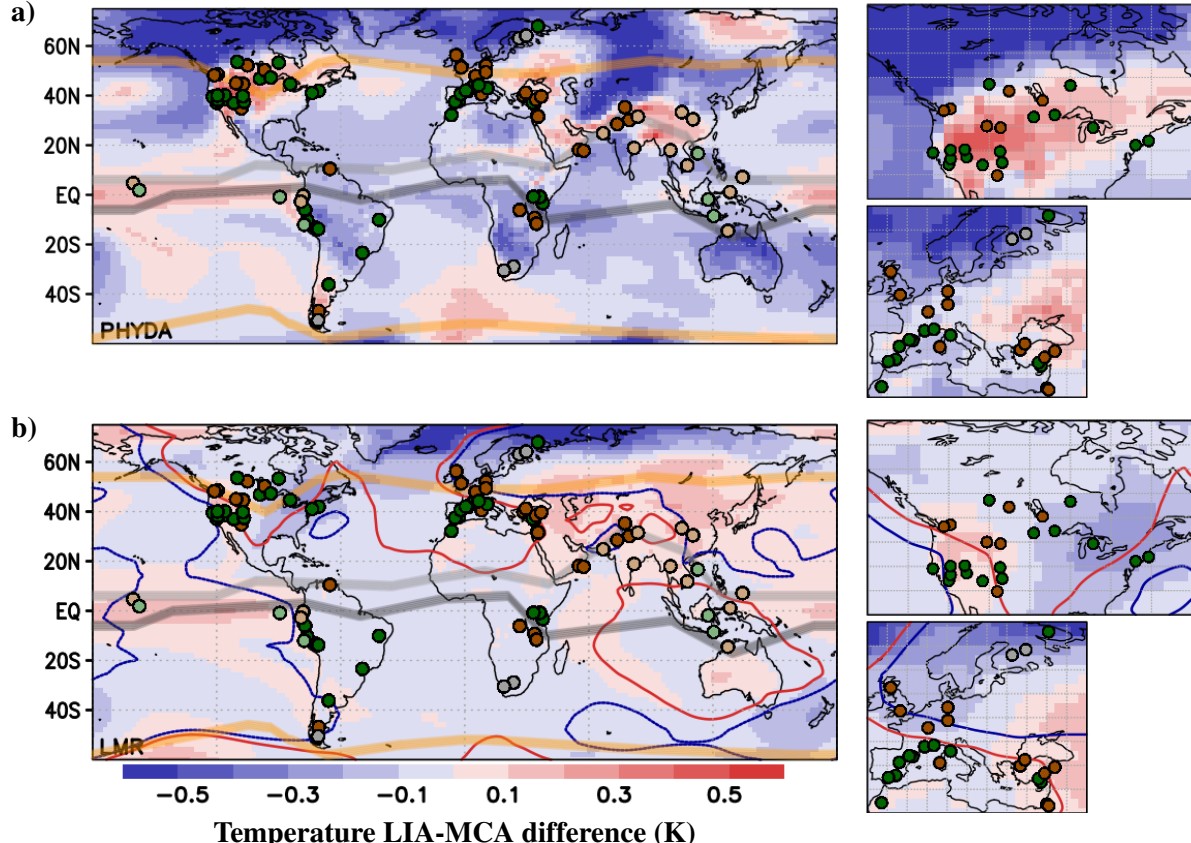

**Figure 3.** LIA-MCA differences in temperature from **(a)** PHYDA, and **(b)** LMR. Dots represent the locations of Fig. 1, with the same color code considered in that figure; positions of the modern ITCZ and NAM and SAM boundaries are also shown, as described in Fig. 1. For the LMR, differences in SLP are also shown with contours of -5 (blue) and 5 (red) Pa. Average of JJA is considered for the PHYDA, while the annual average is considered for the LMR. For each product, regional maps over North America and Europe are shown on the right side of each corresponding panel.

on megadroughts during the MCA period (Cook et al., 2010b; Coats et al., 2013b). For the case of central and western Europe, the PHYDA estimates drier conditions during the LIA, in agreement with the reconstructions of Fig. 1 (Luterbacher et al., 2012), while the DA and LMR estimate wetter conditions. Differences between products are particularly relevant in the Indo-Pacific basin, where the PHYDA and LMR show opposite behavior in Australia, Southeast Asia and Patagonia, and in high latitudes, with opposite behavior in areas of Alaska and northeast Asia. These differences coincide with opposite estimates

in temperature differences as well, with the PHYDA (Fig. 3a) and LMR (Fig. 3b) respectively showing warmer or cooler conditions in Patagonia and cooler or warmer conditions in Australia during the LIA.





**Figure 4.** LIA-MCA differences in PDSI (shading) or precipitation (contours) from the ensemble average of **(a)** CESM-LME, **(b)** CMIP5 and **(c)** CMIP6 LM simulations. Dots represent the locations of Fig. 1, with the same color code considered in that figure; positions of the modern ITCZ and NAM and SAM boundaries are also shown, as described in Fig. 1. For the precipitation, contours of -1 (brown) and 1 (green) mm/month are shown. In all the cases, average of JJA is considered. For each product, regional maps over North America and Europe are shown on the right side of each corresponding panel.







**Figure 5.** LIA-MCA differences in temperature (shading) or SLP (contours) from the ensemble average of **(a)** CESM-LME, **(b)** CMIP5 and **(c)** CMIP6 LM simulations. Dots represent the locations of Fig. 1, with the same color code considered in that figure; positions of the modern ITCZ and NAM and SAM boundaries are also shown, as described in Fig. 1. For the SLP, contours of -5 (blue) and 5 (red) Pa are shown. In all the cases, average of JJA is considered. For each product, regional maps over North America and Europe are shown on the right side of each corresponding panel.

## 3.2 Simulated hydroclimate during the MCA and LIA

The analysis of the model ensembles allows an estimate of those areas that are potentially more impacted by external forcing as estimated by the models, because individual simulations include different realisations of internal variability but share the

external forcing changes imposed as boundary conditions (Fernández-Donado et al., 2013; Jungclaus et al., 2017).



Figure 4 shows the mean PDSI differences between the LIA and MCA for the CESM-LME, CMIP5 and CMIP6 LM ensembles. For the areas of the Middle East and southern Asia the agreement between the DAs, the PHYDA, the LMR and the reconstructions of Fig. 1 extends also to the simulations from CESM-LME (Fig. 4a), CMIP5 (Fig. 4b) and CMIP6 (Fig. 4c). There is also a good agreement in central South America, where the CESM-LME (Fig. 4a), CMIP5 (Fig. 4b) and CMIP6 (Fig. 150 4c) simulations all show a wet LIA for the areas south of the ITCZ, also linked to cooler temperatures during this period (Fig. 5). For northern South America, the dry LIA observed in reconstructions (Fig. 2b,c) is also found, with a limited extension, in CMIP5 (Fig. 4b) and CMIP6 (Fig. 4c) simulations. The agreement found for these areas is consistent with an impact of external forcing on the position of the Atlantic ITCZ (Roldán-Gómez et al., 2022) and the intensity of the monsoon system (Roldán-Gómez et al., 2020).

In western North America and Western Europe, the CMIP5 simulations (Fig. 4b) show a good agreement with the reconstructions of Fig. 1 and with the PHYDA (Fig. 2b). However, this consistency is not so clear in the simulations from CMIP6 (Fig. 4c), for which a dry LIA is found in most regions of North America, associated with limited cooling during that period (Fig. 5c). Disagreement also exists in the CESM-LME (Fig. 4a), which simulates a dry LIA in some areas of southwestern North America and a wet LIA in Western and Northern Europe. These differences can be linked to different atmospheric dy- 160 namics, with the CMIP5 ensemble showing important differences in sea level pressure (SLP) between the MCA and the LIA in most regions of North America and Europe, while the CMIP6 ensemble and the CESM-LME limit these changes to some areas of Southern and Eastern Europe and northern North America. Despite these differences, all the models show relevant changes in these regions during the MCA and LIA, in line with the changes in the North Atlantic Oscillation (NAO) shown in Roldán-Gómez et al. (2020).

In the Indo-Pacific basin, the ensemble average of CESM-LME (Fig. 4a), CMIP5 (Fig. 4b) and CMIP6 (Fig. 4c) LM simulations show small PDSI differences between the MCA and LIA, with no clear agreement with the PHYDA (Fig. 2b) and the LMR (Fig. 2c). The CESM-LME (Fig. 5a) tends to show cooler conditions during the LIA in these regions, but the temperature difference is small compared to that of North America and Europe. For the case of CMIP5 (Fig. 5b) and CMIP6 (Fig. 5c), warmer conditions are found during the LIA for some areas of India and northern Australia. Regarding the atmospheric 170 dynamics, only the LMR shows relevant changes in SLP for the Indo-Pacific, with positive anomalies in the Indian and western Pacific basin and negative anomalies in the eastern Pacific basin during the LIA. This behavior is consistent with a larger impact of internal variability in these areas (Roldán-Gómez et al., 2022), with simulation-dependent changes that are filtered out when working with ensemble averages.

### 3.3 Agreement between products

To assess the agreement between different pairs of products, spatial pattern correlations for the global maps of Fig. 2 and Fig. 4 have been included in Table 4. Despite the regional differences found in areas of North America and the Indo-Pacific basin, the ensembles of CESM-LME, CMIP5 and CMIP6 show global spatial correlations between 0.29 and 0.39. The global correlation of reconstruction-based products is strongly impacted by low correlations in some areas of the southern hemisphere and at high latitudes, with a lower density of proxy data. If these regions are excluded (Table 5), correlations larger than 0.29





**Table 4.** Correlation between the maps of differences LIA-MCA of PDSI from DAs (Fig. 2a), PHYDA (Fig. 2b), LMR (Fig. 2c), CESM-LME (Fig. 4a), CMIP5 (Fig. 4b) and CMIP6 (Fig. 4c). Significant correlations (p < 0.05) accounting for autocorrelation are shown in bold.

|  | PHYDA | LMR | CESM | CMIP5 | CMIP6 |
|---|---|---|---|---|---|
| **DAs** | -0.05 | -0.06 | -0.11 | 0.08 | -0.08 |
| **PHYDA** |  | **0.32** | **0.24** | 0.10 | **0.18** |
| **LMR** |  |  | 0.05 | -0.08 | 0.05 |
| **CESM** |  |  |  | **0.33** | **0.29** |
| **CMIP5** |  |  |  |  | **0.39** |

**Table 5.** Same as Table 4, but considering only the data between 20ºS and 50ºN.

|  | PHYDA | LMR | CESM | CMIP5 | CMIP6 |
|---|---|---|---|---|---|
| **DAs** | **0.35** | -0.08 | -0.04 | **0.30** | 0.08 |
| **PHYDA** |  | **0.29** | **0.35** | 0.11 | **0.27** |
| **LMR** |  |  | **0.24** | -0.10 | 0.16 |
| **CESM** |  |  |  | **0.36** | **0.32** |
| **CMIP5** |  |  |  |  | **0.45** |

are also found between DAs and PHYDA and between PHYDA and LMR. The disagreement between LMR and the other reconstruction-based products in areas of Western Europe and western North America explains the small global correlation between the DAs and LMR. It is important to note that the spatial correlation between reconstruction-based and model-based products is impacted by internal variability. The use of ensemble averages removes the contribution of internal variability as it was manifest in the climate trajectory from the simulated data. For the reconstruction-based products the contribution of

internal variability is not removed, even if the multi-century averages for the LIA and MCA periods emphasize the contribution of the external forcing.

The agreement between sources is particularly poor in areas of the Indo-Pacific basin. As shown in Fig. 6a, there is no clear agreement regarding the sign of the PDSI LIA-MCA differences for areas of Southeast Asia and Australia. There is also no clear agreement for these areas among the individual simulations comprising the CMIP5 and CMIP6 ensembles (Fig. 6b),

suggesting that the hydroclimate of these areas is strongly impacted by simulation-dependent processes, likely those associated with internal variability. These results are in line with the results obtained with the CESM-LME by Roldán-Gómez et al. (2022), showing a relevant impact of internal variability on the position and extension of the Indian and western Pacific ITCZ.

In other areas, most products (Fig. 6a) and most individual simulations (Fig. 6b) show the same sign of PDSI differences such as in central South America, with a drier MCA and a wetter LIA, or the Sahel, extending from Western Africa to the



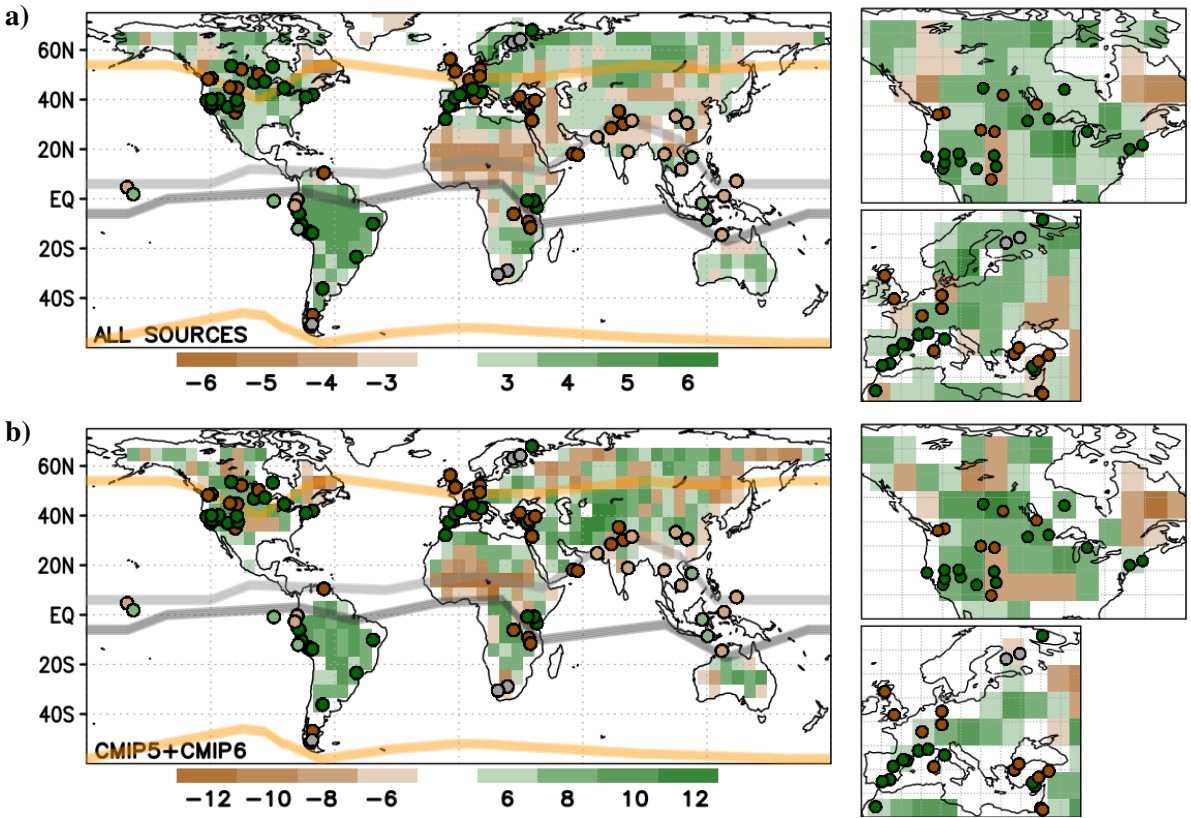

**Figure 6. (a)** Number of products, from DAs, PHYDA, LMR, and the ensemble averages of CESM-LME, CMIP5 and CMIP6, showing positive (green) or negative (brown) PDSI differences LIA-MCA. **(b)** Number of simulations, from the ensembles of CMIP5 and CMIP6, showing positive (green) or negative (brown) PDSI differences LIA-MCA. Dots represent the locations of Fig. 1, with the same color code considered in that figure; positions of the modern ITCZ and NAM and SAM boundaries are also shown, as described in Fig. 1. Regional maps over North America and Europe are shown on the right side of each corresponding panel.

Arabian peninsula, for which most products and simulations show a wetter MCA and a drier LIA. Consistency also extends to areas of western North America, with at least 4 products showing wetter conditions in the south and drier conditions in the north during the LIA, or to northern India, where most products show a wetter MCA and a drier LIA. The agreement for these areas suggest a contribution of external forcing that is more important than that of internal variability.

         The MCA and the LIA are periods extensively addressed in the literature, but they are not the only periods showing relevant
droughts (Cook et al., 2022). The analyses have been then extended to other periods, by extracting time series of PDSI for the whole millennium from the same regions that show coordinated changes during the MCA and LIA (Fig. 7a-e and 8a-e), including western North America, southwestern Europe, northern and central South America, East Africa, Pakistan, India, Southeast Asia and Indonesia. For each region, the correlations between the DAs, the PHYDA, the LMR, and the ensemble average of CESM-LME, CMIP5 and CMIP6 LM simulations have been computed (Fig. 7f-j and 8f-j).







**Figure 7. (a-e)** Time series of PDSI for **(a)** southwestern North America, **(b)** northwestern North America, **(c)** southwestern Europe, **(d)** Pakistan and **(e)** East Africa, obtained from DAs, PHYDA, LMR, and the ensemble average of CESM-LME, CMIP5, and CMIP6 LM simulations. Time series of DAs are not available for those regions not considered in OWDA, NADA, MADA and MXDA. Periods of MCA and LIA are highlighted in red and blue in the middle panels. Dots in the map of the left side represent the locations of Fig. 1, with the same color code considered in that figure. **(f-j)** Correlations between time series of PDSI for each region obtained from DAs, PHYDA, LMR, and the ensemble average of CESM-LME, CMIP5, and CMIP6 LM simulations. Color code for the correlations is the same as included in Fig. 9.





**Figure 8.** Same as Fig. 7 for **(a,f)** Indonesia, **(b,g)** India, **(c,h)** Southeast Asia, **(d,i)** northern South America and **(e,j)** central South America. Time series of DAs are not available for those regions not considered in OWDA, NADA, MADA and MXDA. Time series of CMIP6 are not available for Indonesia, since most CMIP6 simulations do not provide the necessary variables to compute the PDSI for this region.

The correlation between the DAs and PHYDA is significant for western North America, while the correlation between the PHYDA and LMR is significant for central South America, northwestern North America, southwestern Europe, Pakistan, Southeast Asia and Indonesia. Some of these correlations exceed 0.7 and are clearly visible in the time series of PDSI from





these regions. This is the case for the correlation between the DA and the PHYDA in southwestern North America (Fig. 7a,f), and the correlation between PHYDA and LMR in Pakistan (Fig. 7d,i), Indonesia (Fig. 8a,f) and central South America (Fig. 8e,j). Regions showing high correlations between reconstruction-based products are in general those regions showing the largest hydroclimate changes during the LM, like those around the ITCZ (Yan et al., 2015; Anchukaitis and Tierney, 2013), those in the Monsoon region (Fleitmann et al., 2003) and in the area of influence of the NAM (Ortega et al., 2015).

Significant correlations are also found between the ensemble averages of CESM-LME, CMIP5 and CMIP6 for regions of South America, India and Southeast Asia, between CMIP5 and CMIP6 ensembles for areas of southeastern Europe, East Africa and Pakistan, and between CESM-LME and CMIP5 ensembles in areas of southwestern Europe and Indonesia. As for the case of reconstruction-based products, some of these correlations exceed 0.7 and are visible in the time series of PDSI, like those of northern South America (Fig. 8d,i) and Southeast Asia (Fig. 8c,h) for CESM-LME, CMIP5 and CMIP6, and those of East Africa (Fig. 7e,j) and India (Fig. 8b,g) for CMIP5 and CMIP6. In these cases, similarities are found during the whole millennium, including a trend in the 20th century not found in the reconstructions (Ljungqvist et al., 2016). The correlation between simulated products is particularly high for those areas with a larger impact of external forcing, like the Atlantic ITCZ (Roldán-Gómez et al., 2020), consistent with the fact that the use of the ensemble average emphasizes the forcing signal (Fernández-Donado et al., 2013; Roldán-Gómez et al., 2020).

Considering this, the comparison between time series from reconstruction-based products and from model simulations would be only relevant for those areas where the contribution of the forcing dominates. Significant correlations are found between the ensemble average of CESM-LME, CMIP5 and CMIP6 LM simulations and the PHYDA and LMR in northern and central South America and, to a lower extent, in regions of southwestern Europe, northern North America and Southeast Asia, suggesting a relevant contribution of the forcing to the hydroclimate of these areas. Some of these correlations between reconstruction-based products and model simulations exceed 0.5 and are visible in the time series of PDSI, like those in central South America (Fig. 8e,j) and those between the LMR and the ensemble average of CMIP5 in southwestern Europe (Fig. 7c,h). These results can be then linked to a particularly important contribution of the forcing in these regions, present in both proxy data and model simulations.

## 3.4 Correlation between distant regions

As described in the previous sections, distant regions showed similar hydroclimate behaviors during the MCA and LIA. These similarities suggest the presence of coordinated changes between distant regions that go beyond these two periods. To analyse these changes, correlations between the time series of different regions from Fig. 7 and 8 have been computed for the PHYDA (Fig. 9a) and the LMR (Fig. 9b). Both reconstruction-based products show significant positive and negative correlations between distant regions, some of them in agreement with the behavior of reconstructions during the MCA and LIA (Fig. 1).

In particular, both the PHYDA and the LMR show significant negative correlations between central South America and the regions of East Africa and Pakistan. These correlations can be linked to coordinated changes of the Atlantic ITCZ and the monsoon system, in line with those observed in model simulations and associated with shifts of the Atlantic ITCZ in





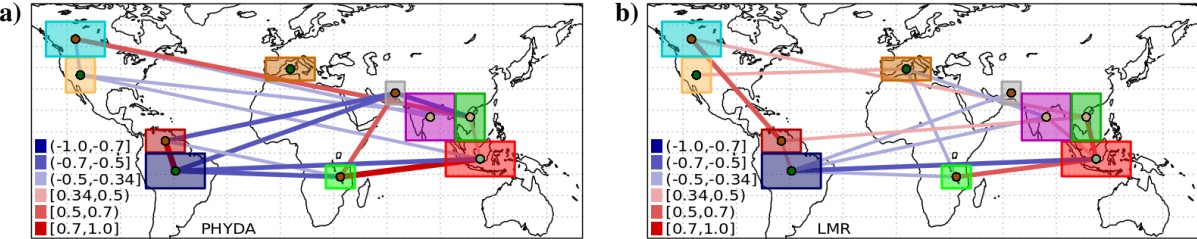

**Figure 9.** Correlations between time series of PDSI for different regions (solid lines), obtained from **(a)** PHYDA and **(b)** LMR. Only significant correlations ($p < 0.05$; $|r| > 0.34$) accounting for autocorrelation are shown, following the color code in the bottom left side of the pannels. The color of the points for each region indicate whether the LIA was wetter (green) or drier (brown) than the MCA in the reconstructions of Fig. 1.

response to changes in the external forcing (Roldán-Gómez et al., 2020, 2022). Both reconstruction-based products also show significant positive correlations across the Indian and Pacific basins, such as those between Indonesia and East Africa, and between Southeast Asia and northwestern North America, consistent with the changes in the Pacific Walker circulation found in simulations (Yan et al., 2015).


The correlations across the Pacific basin are emphasized in the PHYDA, which also shows significant negative correlations between southwestern North America and the regions in India and Indonesia. In contrast, the LMR tends to emphasize the variability in the Atlantic basin, with positive correlations between southwestern North America and southwestern Europe following the pattern of the NAO. The spatial correlations found in the PHYDA and LMR are mainly driven by the model used

for the data assimilation. However, the fact that the PHYDA and LMR respectively ephasize the variability in the Pacific and Atlantic basin shows that the regional variability given by the proxies is consistent with the spatial patterns from the models. This increases the confidence on the ability of the models to reproduce the behavior of the hydroclimate at a global scale, in particular for those areas and those mechanisms more impacted by the external forcing.

## 4 Conclusions

The analysis of different reconstruction-based and model-based products shows consistent coordinated changes during the LM in the hydroclimate of distant regions. The areas more impacted by these changes are those around the ITCZ, like northern and central South America, East Africa and the Indo-Pacific, those in the area of influence of the NAM/NAO, like western North America and Western Europe, and those of the Indian Monsoon region, extending from the Middle East to Southeast Asia. Even if these changes are particularly important during the MCA and the LIA, for which a high number of reconstructions

show drier or wetter conditions, significant correlations between distant regions are also found when considering the whole millennium.

The agreement between reconstruction-based and model-based products for areas of South America and East Africa is indicative of a relevant contribution of the forcing in these regions, consistent with the shifts of the Atlantic ITCZ obtained





in simulations in response to the forcing (Roldán-Gómez et al., 2022). Agreement is also high in the Indian Monsoon region,
consistent with an alteration of the monsoon system as a consequence of changes in the forcing (Roldán-Gómez et al., 2020).
A certain agreement is also found for areas of western North America and Western Europe, consistent with an alteration
of the NAM/NAO in response to changes in the forcing (Roldán-Gómez et al., 2020), even if the different products show
discrepancies at regional scales. These discrepancies may be linked to the contribution of internal variability in these areas, but
also to uncertainties in the reconstructed data, limitations in the physics of the models and inaccuracies in the assessment of
external forcing considered for the simulations. The largest discrepancies between reconstructed and simulated products are
found in the Indo-Pacific basin, suggesting that this area is more impacted by internal variability processes than by changes in
the forcing.

The combination of reconstructions, model simulations and hybrid products is a powerful technique to overcome the limita-
tions of each individual source. On one hand, the sparse hydroclimate records available from proxy data can be completed with
gridded information of atmospheric dynamics and hydroclimate variables provided by model simulations and hybrid products
to extend the analysis to areas not fully covered by the proxies. On the other hand, the representativity of the mechanisms
found in simulations, often involving temperatures, atmospheric dynamics and hydroclimate, like the shifts of the ITCZ, the
alteration of variability modes like the NAM/NAO and the changes in the Walker circulation during the MCA and LIA, could
be confirmed with the hydroclimate information obtained from proxy data and hybrid products.

In this work, we considered an exhaustive and up-to-date compilation of all the available reconstruction-based and model-
based sources to analyse the hydroclimate of the LM at a global scale, with a particular focus on the MCA and LIA as periods
with particular forcing conditions. This exercise, perhaps more typical in the analysis of temperatures but systematically applied
for the first time to the global hydroclimate, provided a novel assessment of global hydroclimate changes. The focus on the
MCA and LIA periods and the comparison with model simulations allowed the isolation of those mechanisms potentially
impacted by the forcing, while the novel approach based on combining reconstruction-based and model-based sources showed
a coherence in the large-scale changes of the hydroclimate of the LM.

## Appendix A: Impact of the definition of the MCA and LIA

For the analyses described in the previous sections, the periods of the MCA and LIA have been respectively defined from 950
to 1250 CE and from 1450 to 1850 CE. This definition may depend on the product and is not applicable at a global scale
(Neukom et al., 2014, 2019). For example, reconstructions of the Indo-Pacific basin (Fig. 1, light brown and light green) tend
to show stable conditions only during the early LIA, until late 1500s CE, rather than during the whole LIA as found in regions
of Europe and North America (Fig. 1, dark brown and dark green). To make the analyses independent from the particular
selection of periods for the MCA and LIA, the differences LIA-MCA of Fig. 2 and Fig. 4 have been replaced in Fig. A1 and
Fig. A2 by the slope of PDSI computed over the the whole period from 950 to 1750 CE. The results obtained with this method
show no major differences with respect to those shown in Fig. 2 and Fig. 4, confirming that the conclusions obtained in the
paper are independent from the particular definition of the MCA and LIA.





**Figure A1.** Slope of PDSI (shading) or precipitation (contours) from **(a)** DAs, **(b)** PHYDA, and **(c)** LMR, obtained with a linear regression for the period 950-1750 CE. Dots represent the locations of Fig. 1, with the same color code considered in that figure; positions of the modern ITCZ and NAM and SAM boundaries are also shown, as described in Fig. 1. For the slope of precipitation, contours of -0.001 (brown) and 0.001 (green) mm/month/year are shown. Average of JJA is considered for the PHYDA and the DAs, while the annual average is considered for the LMR. For the PHYDA and the LMR, regional maps over North America and Europe are shown on the right side of each corresponding panel.

## Appendix B: Seasonal changes in the PDSI

In the results presented in the previous sections, the JJA PDSI values from the DAs, the PHYDA and the simulations from CESM-LME, CMIP5 and CMIP6 are compared with the annual PDSI provided by the LMR. To confirm that the use of





**Figure A2.** Slope of PDSI (shading) or precipitation (contours) from the ensemble average of **(a)** CESM-LME, **(b)** CMIP5 and **(c)** CMIP6 LM simulations, obtained with a linear regression for the period 950-1750 CE. Dots represent the locations of Fig. 1, with the same color code considered in that figure; positions of the modern ITCZ and NAM and SAM boundaries are also shown, as described in Fig. 1. For the slope of precipitation, contours of -0.001 (brown) and 0.001 (green) mm/month/year are shown. In all the cases, average of JJA is considered. For each product, regional maps over North America and Europe are shown on the right side of each corresponding panel.



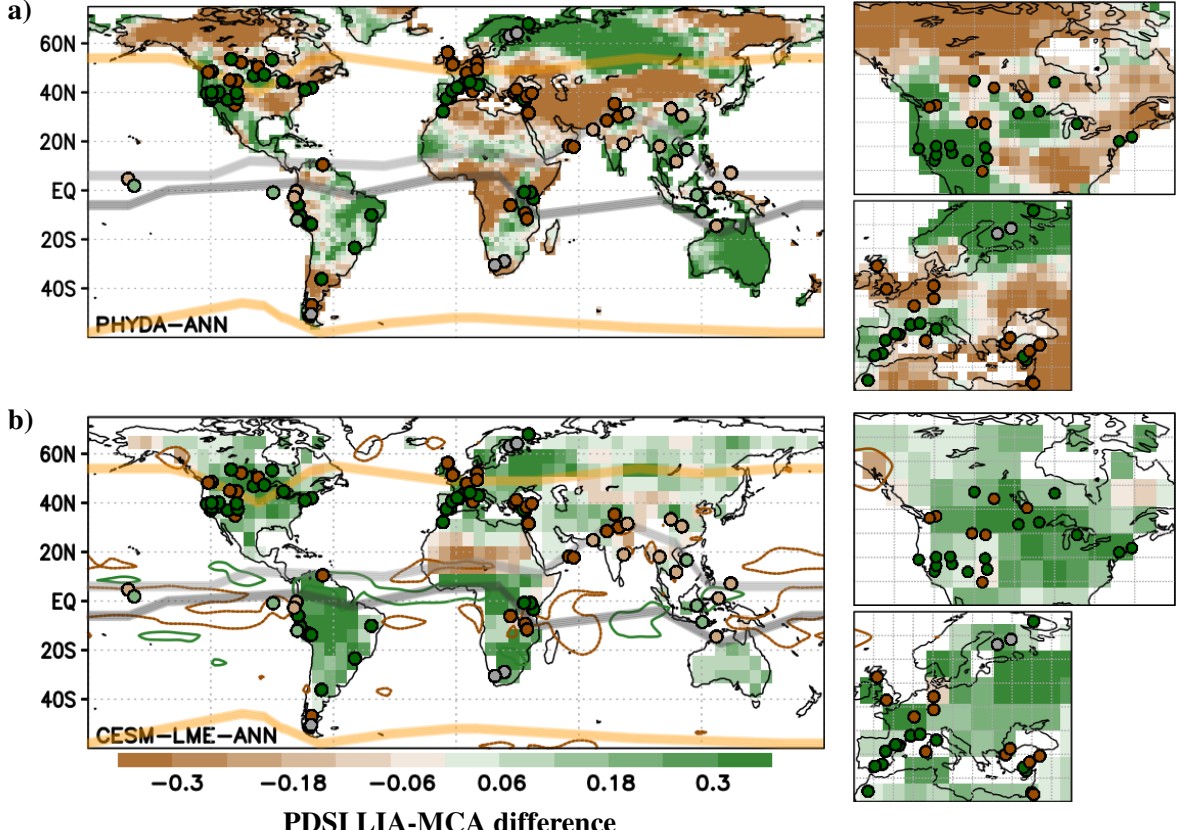

**Figure B1.** LIA-MCA differences in PDSI (shading) or precipitation (contours) from **(a)** PHYDA and **(b)** the ensemble average of CESM-LME, considering annual averages. Dots represent the locations of Fig. 1, with the same color code considered in that figure; positions of the modern ITCZ and NAM and SAM boundaries are also shown, as described in Fig. 1. For the precipitation, contours of -1 (brown) and 1 (green) mm/month are shown. For each case, regional maps over North America and Europe are shown on the right side of each corresponding panel.

different averaging periods does not have a significant impact on the results, the annual PDSI from the PHYDA and the different ensembles of simulations have been also analysed. The difference between the LIA and MCA for the annual PDSI of the PHYDA and the ensembles of CESM-LME, CMIP5 and CMIP6 is respectively shown in Fig. B1a, Fig. B1b, Fig. B2a and Fig. B2b. When comparing these results with those obtained with the average of JJA (Fig. 2b and Fig. 4) only minor differences are observed, mostly limited to regional scales. This shows that despite the relevant seasonal changes found in the atmospheric

variables, the cumulative behavior of the PDSI makes it less sensitive to the season.



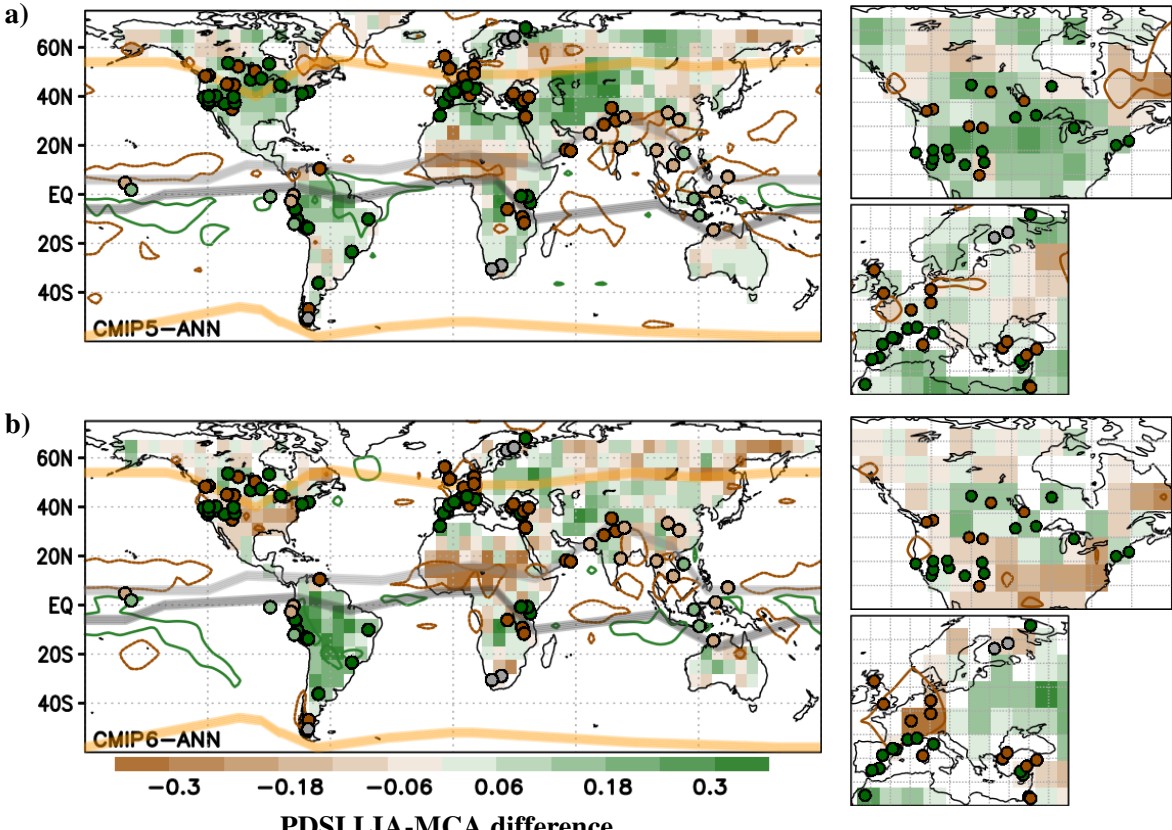

**Figure B2.** LIA-MCA differences in PDSI (shading) or precipitation (contours) from the ensemble average of **(a)** CMIP5 and **(b)** CMIP6 LM simulations, considering annual averages. Dots represent the locations of Fig. 1, with the same color code considered in that figure; positions of the modern ITCZ and NAM and SAM boundaries are also shown, as described in Fig. 1. For the precipitation, contours of -1 (brown) and 1 (green) mm/month are shown. For each case, regional maps over North America and Europe are shown on the right side of each corresponding panel.

*Author contributions.* This study is part of PJRG's PhD. PJRG contributed with data processing, analysis of results and writing of the paper. JFGR, JES and FGP contributed to the analysis and discussion of results and to writing the paper.

*Competing interests.* The authors declare that they have no conflict of interest.



*Acknowledgements.* We gratefully acknowledge the SMILEME (PID2021-126696OB-C21) project. JES was supported in part by the US

National Science Foundation grants OISE-1743738 and AGS-2101214. FGP was funded by a FPI contract (PRE2019-090694) of the Spanish
Ministry of Science and Innovation.



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
