# Peer review of "Model and proxy evidence for coordinated changes in the hydroclimate of distant regions over the Last Millennium"

_Climate of the Past, 2023_

## Author Comment (AC1)

**Responses to reviewer's comments for "Model and proxy evidence for coordinated changes in the hydroclimate of distant regions over the Last Millennium"**

We are grateful to the reviewers for their comments and suggestions, all of which have been helpful for improving the manuscript. We respond to each of the comments in our thorough replies below, providing in gray the comments from each review and in black our responses. Line and figure numbers correspond to the lines and figures in the newly revised manuscript unless otherwise noted.

**Community (Matthew Kirby):**

**C1C1**
Might add other coastal sw US sites -

Zaca Lake (Kirby et al., 2014) - Wet LIA, Dry MCA

Lake Elsinore (Kirby et al., 2019) - Wet LIA, Dry MCA

Abbott Lake (Hiner et al., 2016) - Wet LIA, Dry MCA

Maddox Lake (Kirby et al., in press in Quaternary Research) - Wet LIA, Dry MCA

These sites will extend your analysis into the coastal sw US as well as far northwest CA (Maddox Lake)

The suggested references have been added to Fig. 1 and Table 2.

**Reviewer 1:**

Review of "Model and proxy evidence for coordinated changes in the hydroclimate of distant regions over the Last Millennium" by Roldán-Gómez et al.

The authors present a comprehensive overview of existing hydro-climatic proxies, reconstructions and simulations and draw links to circulations changes such as shifts of the ITCZ. I suggest to publish this study after major revisions.

**R1C1**
On the one hand, I think the manuscript could be condensed by removing the analysis of temperature to focus more on the hydroclimate.

The analysis of temperature and atmospheric dynamics, including Fig. C1 and Fig. C2, has been moved to Appendix C. In the main text, only a reference to the Appendix has been kept (P14 L 204: "As shown in Appendix C, this disagreement can be explained by a different behavior in terms of temperatures and atmospheric dynamics").

**R1C2**
On the other hand, I suggest to add more details: The authors check if reconstructions for different seasons show comparable variability. However, I suggest a more extensive discussion of seasonality issues. Many northern hemisphere proxies/reconstructions may be biased towards JJA, southern hemisphere proxies towards DJF and tropical proxies toward one or two rain seasons. Simulations could give us an idea how much e.g. missing winter precipitation information in the extratropics

may influence the results of this study.

Refer to R1C8 and R1C12 for the discussion on the seasonality of the proxies.

**R1C3**
It is also not clear to me if the proxy collection is independent from the proxies used in the reconstructions?

The proxies included in the references of Fig. 1 and Table 2 are not necessarily independent from the proxies used in the generation of the reconstructed products. Some of them are used while some others are not. The goal of the Fig. 1 and Table 2 is not to create a new data set of proxies independent from the existing ones, but to analyse the literature based on proxies to identify the areas showing wetter or drier conditions during the MCA and LIA.

As per comment R2C3, Fig. 1 has been moved to a dedicated section and more details on the construction of this figure has been included.

**R1C4**
However, my main concern after seeing the time series in Fig. 7 are possible problems with the PDSI values due to different trends in the simulations, especially in the 20th century. Will the PDSI differences between CESM, CMIP5 and CMIP diminish if PDSI is calculated and calibrated during the preindustrial period? At least the y-axes scales in Fig. 7 suggest that for DAs and PHYDA PDSI varies between positive and negative values while it appears to be mostly negative in the simulations during the pre-industrial period. Does this effect the LIA-MCA differences, too?
In general it would be helpful understand how PDSI calculations based on the simulations could be influenced by model biases of temperature and precipitation on which PDSI is based. Therefore, I would suggest to compare PDSI of all reconstructions and simulations to instrumental observation based PDSI in the 20th century, e.g. the CRU PDSI data set (https://crudata.uea.ac.uk/cru/data/drought/).

The impact of the calibration period in the computation of the scPDSI was analysed, and it was found to be very limited and not impacting the overall results of the analysis. As an example, Fig. R1 shows the scPDSI time series obtained from one particular simulation (first simulation of CESM-LME) for two grid points in regions showing scPDSI biased to negative values (Spain and Pakistan). The same time series have been computed using the whole millennium as calibration period and limiting the calibration to the pre-industrial period, without major differences in the results.

As suggested in the comment, the PDSI from the CRU data set has been included in Fig. 6 and 7, and the text has been updated accordingly (P10 L137: "For the 20th Century, the PDSI from the different sources have been compared to the instrumental observation based PDSI from the Climatic Research Unit (CRU; Barichivich et al., 2021).")

For the locations in India, Pakistan, East Africa and western North America, the CRU data set

shows negative values of the PDSI, in line with the values shown in the simulations. For the locations in southwestern Europe, Indonesia, Southeast Asia and South America, the PDSI values from the CRU data set are zero-centered, more in line with the values from DAs and PHYDA. This shows that at least part of the biases observed in the PDSI from the simulations could correspond to the real biases for those regions.

[Figure]

**Figure R1.** scPDSI obtained from the first simulation of CESM-LME for two grid points, in **(a)** Spain and **(b)** Pakistan, based on a calibration period 850-2000 CE (LM) and 850-1850 CE (PRE).

**R1C5**
Abstract:
L. 1ff, l. 8 ff. Please try to shorten your often really long sentences. Especially with all the references, sentences are often complicated to read on the first pages.

Sentences have been shortened in the abstract and in the introduction.

In particular, the first lines of the abstract have been split in shorter sentences (P1 L1: "The Medieval Climate Anomaly (MCA; ca. 950-1250 CE) and the Little Ice Age (LIA; ca. 1450-1850 CE) were periods generally characterised by respectively higher and lower temperatures in many regions. However, they have also been associated with drier and wetter conditions in areas around the Intertropical Convergence Zone (ITCZ), the Asian Monsoon region, and in areas impacted by large-scale climatic modes like the Northern and Southern Annular Modes (NAM and SAM,

respectively). To analyse coordinated changes in large-scale hydroclimate patterns, and whether similar changes also extend to other periods of the Last Millennium (LM) outside the MCA and the LIA, reconstruction-based products have been analysed. This includes the collection of tree-ring based Drought Atlases (DA), the Paleo Hydrodynamics Data Assimilation product (PHYDA) and the Last Millennium Reanalysis (LMR).")

In the introduction, the following sentences have been split:
- P2 L33: "For example, reconstructions from southwestern North America (Graham et al., 2007; Meko et al., 2001; Cook et al., 2004, 2010b; Hughes and Funkhouser, 1998; Anderson, 2011) and the Mediterranean basin (Luterbacher et al., 2012; Martín-Puertas et al., 2010; Morellón et al., 2009) show drier conditions during the MCA and wetter conditions during the LIA. On the contrary, reconstructions from northwestern North America northwestern North America (Steinman et al., 2013; Stevens and Dean, 2008), central Europe (Büntgen et al., 2010, 2011) and the British Isles (Wilson et al., 2012; Proctor et al., 2000) tend to show wetter conditions during the MCA and drier conditions during the LIA."
- P2 L40: "An antiphased relationship between the MCA and LIA is also observed in tropical areas of South America. Wetter conditions are found during the MCA in areas of northern South America, like the Cariaco Basin in Venezuela (Haug et al., 2001), and drier conditions in areas of central South America, like Peru (Bird et al., 2011; Reuter et al., 2009; Vuille et al., 2012; Thompson et al., 1986; Apaéstegui et al., 2014) and Eastern Brazil (Vuille et al., 2012; Taylor, 2010; Novello et al., 2012)."
- P2 L48: "Changes in the ITCZ have been also associated with coordinated changes in the hydroclimate of the Indian Monsoon region. Wetter conditions during the MCA and drier conditions during the LIA are found  for wide areas of Pakistan (von Rad et al., 1999; Treydte et al., 2006), India (Sinha et al., 2007; Berkelhammer et al., 2010; Bryson and Swain, 1981; Phadtare and Pant, 2006) and the Arabian Sea (Burns et al., 2002; Fleitmann et al., 2003; Anderson et al., 2002; Gupta et al., 2003)."
- P2 L52: "Alterations of the ITCZ have been also associated with changes in the hydroclimate of the Indo-Pacific basin (Atwood et al., 2021). A marked transition between the MCA and LIA is found in areas of China (Wang et al., 2005; Zhang et al., 2008; Hu et al., 2008), Southeast Asia (Buckley et al., 2010, 2007), Indonesia (Griffiths et al., 2016; Newton et al., 2006; Tierney et al., 2010b), northern Australia (Denniston et al., 2016) and the eastern Pacific islands (Sachs et al., 2009; Higley et al., 2018)."

**R1C6**
L. 12: Please explain how internal variability could cause a 300 to 400 year warm or cold period?

Even if the impact on temperature is generally in the short term, here we refer to hydroclimate changes. The internal variability can impact on the hydroclimate on multi-centenial scales, for example with shifts of the ITCZ, as shown by Roldán-Gómez et al. (2022).

**R1C7**
L. 14: Please clarify the last sentence of the abstract.

We are identifying the areas with a larger impact of internal variability. For these areas, the

evolution of the hydroclimate provided by model simulations will depend on the initial conditions, and will be not necessarily the same as in the real world. On the contrary, for the areas more impacted by the external forcing, if the forcing assessment used in the simulations is accurate enough, one could expect that the model simulations are able to reproduce the real evolution of hydroclimate.

The sentence has been modified to clarify this (P1 L14: "These results may help to identify not only those areas showing coordinated changes, but also those regions more impacted by the internal variability, where forced model simulations would not be expected to successfully reproduce the evolution of past actual hydroclimate changes.").

**R1C8**
Introduction:
This is a great global collection of proxies. I miss information and discussion on the seasonality of the proxies, which could be added to Tab.1. Could seasonality explain part of the differences between locations?

Seasons covered by the proxies have been included in Table 2, and, for comparison, seasons provided by the different products have been also included in Table 3. The text has been updated accordingly (P10 L140: "The analysis of hydroclimate based on proxy data is sensitive to the seasonality of the proxies. As shown in Table 2, most proxies provide annual values, but certain proxies of the Northern Hemisphere are linked to the hydroclimate of the boreal summer (June, July and August; JJA), while certain proxies in the Southern Hemisphere are associated with the boreal winter (December, January, and February; DJF). The selection of the season considered in the different products is then a critical point for the analyses. As shown in Table 3, the PHYDA provides water-year annual averages and seasonal averages for JJA and DJF. The LMR only provides annual averages, while the Northern Hemisphere DAs exclusively target the JJA period. For all the analyses, JJA values are considered for all the products except for the LMR, for which the annual values are used. To confirm that the annual results from the LMR can be meaningfully compared to the seasonal results from the other products, the annual PDSI from the PHYDA and the ensembles of CMIP5, CMIP6 and CESM-LME are analysed in Appendix B.")

Looking at the seasons of the proxies in Table 2 and at the behavior during the MCA and LIA in Fig. 1, no obvious correlation between the seasonaility of the proxy and the long-term behavior is found. The difference between locations looks more linked to the location itself than to the seasonality of the proxy.

Regarding the model simulations, the impact of the seasonality is assessed in Appendix B, concluding that it is not relevant for the conclusions (P22 L351: "When comparing these results with those obtained with the average of JJA (Fig. 3b and Fig. 4) only minor differences are observed, mostly limited to regional scales. This shows that despite the relevant seasonal changes found in the atmospheric variables, the cumulative behavior of the PDSI makes it less sensitive to the season.")

**R1C9**

Fig. 1: All arrows are difficult to understand in the map. Additionally, all codes can only be well understood after scrolling two pages down to Tab. 2. Therefore, Fig.1 and Tab.2 should be right next to each other. Maybe better use the abbreviations like NAO, ITCZ, etc. directly in the map directly instead of A1, B4, ...

Fig. 1 has been updated to replace the codes by abbreviations, and Table 2 has been put as Table 1, immediately after the Fig. 1.

**R1C10**

Data and methods:
How could the much small number of proxies for the MCA compared to the LIA influence all proxy-based data sets and therefore the results of this study?

A new figure (Fig. 2) has been added, including the location of the proxies assimilated by the PHYDA and LMR, and whether they extend to the MCA and LIA. A remark has been added in the methods section, to clarify that for certain regions the PHYDA and LMR may be mostly defined by the model (P4 L99: "Even if the PHYDA and LMR provide data for the whole millennium, some of the proxies used for the assimilation do not extend so far in the past. Figure 2 shows the distribution of proxies from Steiger et al. (2018a), including those that cover the MCA and LIA. There is a large amount of proxies covering these periods in regions like North America and Europe, so for these regions the products would be expected to follow the behavior of the proxies. However, for other regions like central Africa and eastern South America, the lack of proxies could increase the contribution of the model used for the assimilation, and thus the uncertainty of the products.").

**R1C11**

Are all data sets and simulations going back to 950 C.E.? In Tab. 3 I see the simulations of Tett et al. 2007 listed, which do not go that far back in time as fr as I know. The MXDA (Stahle et al. 2016) is also does not cover the MCA and you also do not show it in Fig. 2. Which DAs are really included in all analysis including the correlation analysis in Tab. 4?

The starting year has been included in Table 3. All the products go back to 950 CE, except for the MADA. The MXDA has been removed from the table, since it is not used for any of the figures of the paper, and a clarification on the use of the MADA has been added in the text (P4 L97: "Due to the fact that they do not extend back into the MCA and LIA, other DAs have not been included in our analyses, and since the MADA does not extend back into the MCA, it has been only considered for the analysis of time series of Sect. 3.3.").

Regarding the simulations of HadCM, the reference to Tett et al. (2007) was not the correct one. We are using the simulations performed for the CMIP5/PMIP3, which start in 850 CE. The correct reference (Schurer et al., 2014) has been put instead.

**R1C12**

L. 108: You say that you focus on the JJA season. Is the analysis only done for the northern

hemisphere? Because in the southern hemisphere the reconstructions are possible representing rather the DJF season.

The analysis is done both for NH and SH, even if the focus is put on the NH and then in the JJA season. The choice of JJA for the simulated data and the data from the PHYDA was mostly intended to allow for a more relevant comparison with the data from the DAs, which is only provided for JJA. However, the LMR is provided annually, and some of the proxies in Fig. 1 represent mostly annual values or the DJF season (especially for those in the SH). As described in R1C8, the seasons covered by the proxies have been included in Table 2, and the seasons provided by the different products have been also included in Table 3.

To verify that this choice is not significantly impacting the results, the analyses have been repeated with annual values in Appendix B.

**R1C13**

3.1 Reconstructions
Fig. 2: Surprisingly large disagreement between drought atlases and PHYDA even in the sign of change for western half of Europe and parts of North America with fewer proxies. This should be mentioned in the results more clearly. Is this a result of the different methods or the underlying data? For the proxy locations indicated by the dots it is unclear to me, which are independent/additional information and which are already included in which of the gridded reconstructions and therefore may indicate better agreement.

Discrepancies in western Europe and central and eastern North America have been highlighted in the text (P13 L173: "For the case of central and western Europe, the PHYDA estimates drier conditions during the LIA, in agreement with the reconstructions of Fig. 1, while the DA and LMR estimate wetter conditions. The same happens for areas of central and eastern North America, especially for those areas with a small number of proxies covering the MCA (Fig. 2).").

The discrepancies between DAs and PHYDA may come both from the methods and from the selection of proxies, but very likely most of the differences are linked to the methods. Even if the PHYDA uses multiple types of proxies and the DAs are only based on tree rings, most of the records used in the DAs are also included in the PHYDA, so the selection of proxies is not strongly different for both products. The fact that discrepancies mostly appear in the areas with less proxies leads to think that this is linked to the different way these areas are "filled", with data assimilation in the PHYDA and with interpolation in the DAs.

Refer to comment R1C3 for the discussion on the independence of the proxies of Fig. 1.

**R1C14**

Fig. 3 and 5: Does it even make sense to present temperature reconstructions or even the temperature focussed LMR data set in this study on hydroclimate? I would suggest to remove these figures and related text or move it to the appendix.

Refer to comment R1C1 for the analysis of temperatures.
Refer to comment R1C25 for the use of LMR.

**R1C15**

3.2 Simulated hydroclimate
Yes, model ensembles should let us understand the forced signal. However, there is surprisingly large disagreement between model generations/sets. How different are CMIP5/6 PMIP3/4 forcings averaged of MCA and LIA? Or is the reason rather that different models are in both sets?

Even if not shown in the paper, analyses for the individual simulations of CMIP5 and CMIP6 ensembles have been also performed. These individual analyses allow to compare the same model with the forcings of CMIP5 and CMIP6. In Fig. R2, the MCA-LIA differences are presented for the CMIP5 and CMIP6 simulations of MPI and MRI. When comparing these maps, one can see similarities between the MPI and MRI simulations of CMIP5 (for example in areas of central and eastern Asia and western North America) and between the MPI and MRI simulations of CMIP6 (for example in areas of eastern North America and southern and eastern Africa), suggesting a relevant impact of the forcing specifications. However, there are also commonalities between simulations of MPI of CMIP5 and CMIP6 (for example in central Europe, Southeast Asia and southern South America) and between simulations of MRI of CMIP5 and CMIP6 (for example in India and western North America), suggesting also a relevant impact of the model.

When analysing the ensemble averages of CMIP5 and CMIP6, the differences can be then explained both by the use of different forcing specifications and by the use of a different set of models. To have a more detailed insight on this, more specific studies would be needed.

[Figure]

a) **CMIP5-MPI**  b) **CMIP6-MPI**

c) **CMIP5-MRI**  d) **CMIP6-MRI**

[Figure]

[Figure]

**Figure R2.** MCA-LIA differences in PDSI from the **(a)** CMIP5 MPI, **(b)** CMIP6 MPI, **(c)** CMIP5 MRI and **(d)** CMIP6 MRI simulations.

**R1C16**

L. 155: I would not call this a good agreement over western North America. In contrast to the reconstruction especially the simulations CESM-LME show a continent wide uniformly more moist LIA in North America and Europe.

Text has been modified to clarify that the extension of the wet LIA is more limited in the reconstructions (P14 L194: "The CMIP5 simulations (Fig. 4b) show a wetter LIA in most areas of western North America and Western Europe, which is also found with a more limited extension in the reconstructions of Fig. 1 and the PHYDA (Fig. 3b). However, this consistency is not so clear in the simulations from CMIP6 (Fig. 4c), for which a dry LIA is found in most regions of North America.")

**R1C17**

3.3. Agreement

L. 182: You mention the role of internal variability. Can you explain why multi-century averages in the case of reconstructions should still contain internal variability? Can this be shown in control simulations?

Refer to comment R1C6 for the discussion on the role of internal variability.

Indeed the use of multi-century averages limits the impact of internal variability, as already mentioned in the text (P16 L225: "For the reconstruction-based products the contribution of internal variability is not removed, even if the multi-century averages for the LIA and MCA periods emphasize the contribution of the external forcing"), but the internal variability can also have an impact on larger time scales. This is the case for example of the shifts of the ITCZ analysed by Roldán-Gómez et al. (2022).

**R1C18**

You mention the role of the ITCZ for the Indo-Pacific region but why is the disagreement large in the rather data rich regions of western North America and western Europe?

For the model simulations, the differences in the hydroclimate of regions of western North America and western Europe seem linked to differences in the atmospheric dynamics, as explained in the text (P14 L198: "Disagreement also exists in the CESM-LME (Fig. 4a), which simulates a dry LIA in

some areas of southwestern North America and a wet LIA in Western and Northern Europe. As shown in Appendix C, this disagreement can be explained by a different behavior in terms of temperatures and atmospheric dynamics.").

A more detailed analysis has been included in the new Appendix C:

"The drier LIA shown by CMIP6 simulations for regions of North America (Fig. 4c) could be also associated with a limited cooling during that period in the CMIP6 ensemble (Fig. C2c) compared to that shown in CMIP5 (Fig. C2b) and CESM-LME (Fig. C2a), while the wetter LIA shown by CESM-LME simulations in Western and Northern Europe (Fig. 4a) could be associated with the fact that CESM-LME shows cooler conditions during the LIA for these regions (Fig. C2a) compared to those simulated in CMIP5 (Fig. C2b) and CMIP6 (Fig. C2c) ensembles.

These differences can be linked to differences in atmospheric dynamics, with the CMIP5 ensemble showing important differences in sea level pressure (SLP) between the MCA and the LIA in most regions of North America and Europe (Fig. C2b), while the CMIP6 ensemble (Fig. C2c) and the CESM-LME (Fig. C2a) limit these changes to some areas of Southern and Eastern Europe and northern North America."

**R1C19**

Fig. 6: I generally like the idea of showing the signal in all products together. However, this seem to be a rather subjective weighting of the data sets. Why is LMR included although it is not really good in reconstructing drought? Why are state-of-art CMIP6 simulations weighted the same as older simulations? In b) there should be 8 CMIP5 and only 3 CMIP simulations according to Tab.3. Is the signal between CMIP5 and CMIP6 simulations just different because we look at different models or because of newer model versions or because of different forcings? How sensitive are all results to changes in the weighting?

The goal of Fig. 5 (previous Fig. 6) is to synthesise the results, but individual analysis for each product are also available in Fig. 3 and 4. There is indeed a subjective choice in the weighting, but it was preferred to assign the same weight to each product/ensemble, because we have no particular arguments to assign a different weight to each product, and mostly to keep the link with the maps of Fig. 3 and 4, in which each product/ensemble is shown in a separate map.

The figure 5b considers the number of simulations and not the number of models. There are then 10 simulations of CMIP5 and 4 simulations of CMIP6.

Refer to comment R1C15 for the differences between CMIP5 and CMIP6 and the contribution of the different models and the different specifications of external forcing.

**R1C20**

Fig. 7: Now for Pakistan is becomes clear that there in no DA data for the MCA. I guess this just needs to be clarified in the data and methods section that MADA is only used in this part of the study and not for the LIA minus MCA comparison.

A clarification has been added in the methods section (P4 L97: "Due to the fact that they do not

extend back into the MCA and LIA, other DAs have not been included in our analyses, and since the MADA does not extend back into the MCA, it has been only considered for the analysis of time series of Sect. 3.3.")

**R1C21**
Surprising how small the agreement between DAs and PHYDA is outside of western North America although they are probably based on similar tree-ring collections. This could be mentioned more prominently because it is probably worth knowing for many people working in the field.

Discrepancies in western Europe and central and eastern North America have been highlighted in the text (P13 L173: "For the case of central and western Europe, the PHYDA estimates drier conditions during the LIA, in agreement with the reconstructions of Fig. 1, while the DA and LMR estimate wetter conditions. The same happens for areas of central and eastern North America, especially for those areas with a small number of proxies covering the MCA (Fig. 2).").

Refer to comment R1C13 for more detailed comments on the discrepancies between DAs and PHYDA in central and eastern North America and western Europe.

**R1C22**
Correlations are probably affected by 20th century trend in some simulations. I suggest calculating PDSI in the pre-industrial period and showing correlations for this period, too. This may effect your discussion on the role of internal vs external variability as well.

Correlations for the pre-industrial period have been added to Fig. 6 and 7. The correlations are smaller when using only the pre-industrial period, but still significant in most of the cases, so the conclusions extracted from this analysis remain valid.

The text have been modified to clarify this (P17 L261: "In these cases, similarities are found during the whole millennium, including a trend in the 20th century not found in the reconstructions (Ljungqvist et al., 2016). If only the pre-industrial period is considered, the correlations are smaller but still significant for most of these areas. The correlation between simulated products is particularly high for those areas with a larger impact of external forcing, like the Atlantic ITCZ (Roldán-Gómez et al., 2020), consistent with the fact that the use of the ensemble average emphasizes the forcing signal (Fernández-Donado et al., 2013; Roldán-Gómez et al., 2020).")

**R1C23**
Concerning the role of internal vs external variability, I would also appreciate a discussion of how the data sparse region with less reconstruction skill can have an influence.

An increase in the coverage of proxies could improve the modelling of internal variability, and then the agreement between spatial patterns of models and reconstructed products. In areas with more proxies more likely the reconstruction-based and model-based products will reproduce the internal variability, and then a better agreement would be expected between them. In areas with less proxies, the response to external forcing and internal variability shown by the simulations may differ more

from the actual response, since they would be based on a covariance structure driven by the proxy-rich regions.

**R1C24**

3.4. Correlation between distant regions
How much of the teleconnections discussed here are really constrained by the proxy data? The reconstructions methods use a stationary covariance structure of the underlying model simulations and in some of the regions discussed here, there is very little data assimilated. Are similar teleconnections already existing in the simulation ensembles? In this regard, it would be helpful to see the location where data has been assimilated into PHYDA/LMR already since the beginning of the MCA.

Refer to comment R1C10 for the proxy data assimilated into PHYDA/LMR.
Refer to comment R1C25 for the link of teleconnections to proxy data and to the underlying model simulations.

**R1C25**

I would also not show the LMR as it did not seen to do a great job as the drought reconstruction. Rather show the teleconnections in the simulations, especially CESM with underlies PHYDA if I remember correctly.

Even if the LMR is more focused on temperature than on drought, we consider that the results of the LMR may be relevant for future studies based on this product, so we prefer to keep them.

To analyse whether the teleconnections from PHYDA come from the CESM or from the proxies, the same correlation map has been added for the CESM-LME (Fig. 8c). The correlations obtained only with the CESM-LME differ from those of the PHYDA more than those from the LMR, showing that the contribution of the proxies to the teleconnections is more important than that of the model.

The text has been modified to comment on this (P21 L293: "The spatial correlations found in the PHYDA and LMR are mainly driven by the model used for the data assimilation. However, if the correlations are computed for the CESM-LME (Fig. 8c) without any data assimilation, only positive correlations associated to the impact of temperature in the PDSI are obtained").

**Reviewer 2:**

This paper takes a detailed look at available proxy evidence for hydroclimate changes between the medieval warm period and little ice age and compares that to the equivalent change in a large number of different model simulations. I think that this paper is an important and interesting contribution to the literature and should be published after a few major points are addressed. I hope my suggestions are helpful.

Major comments
**R2C1**
The first is that more detail should be added about the use of PDSI as a variable to study and the

implications of looking at this. Perhaps in the introduction. What is PDSI? What does it represent and do model simulations do a good job at representing this?

[revised manuscript text omitted]

Even if it is possible to identify those areas with a smaller uncertainty and a larger contribution of external forcing from the methods section, we prefer to keep the global analyses in the subsequent discussion. The areas impacted by larger uncertainties and a larger contribution of internal variability still have a certain amount of proxies and a certain contribution of the forcing, that could still provide meaningful results.

Refer to R1C10 and R1C23 for more details on the impact of proxy scarcity and internal variability.

**R2C3**
The inclusion of figure 1 in the introduction seemed a bit strange to me. As it is not clear (at least to me) what is and what is not original work. Is this figure essentially a reproduction of previous work or is it a new compilation? It looks like an impressive amount of work to me (although I am no proxy expert) so I would have thought it could have its own section. Equally for such an important figure there seems to be no discussion of how the proxies were selected – what does "reporting changes from wetter changes from wetter to drier or from drier to wetter" mean? Are they known to represent changes in PDSI? What seasons are they sensitive to? Do they span the full MCA and LIA periods? This information is important if they are going to be subsequently compared to other proxy and model products. I think removing this to its own section could also help improve the introduction section.

Figure 1 presents a new compilation of references. As suggested, the figure has been moved to a separate section (section 2, "Evidences from the literature") to highlight that this compilation is an original work.

In this new section, a more detailed description of what the proxies represent has been added:

"To survey the available evidence, we assessed 96 reconstructions reporting changes from wetter to drier or from drier to wetter conditions during the transition from the MCA to LIA (Fig. 1, with detailed references in Table 2). The reconstructions used in this compilation include proxy data from tree rings, marine and lake sediments, speleothems, ice cores and documentary information. These sources provide information about precipitation, moisture, level of lakes and river flows, which can be linked to drier and wetter conditions. Some reconstructions represent the annual hydroclimate while some others are sensitive to a particular season, as described in Table 2.

All the reconstructions included in the compilation cover the MCA and LIA. Some of them show consistent drier or wetter conditions during the whole LIA (dark green and dark brown), while some others show drier or wetter conditions until late 1500s CE, during the early LIA (light green and light brown), as reported for areas of Pakistan, western India and southern China by Graham et al. (2010).

The reconstructions from Fig. 1 are used as reference for the analysis of reconstruction-based and model-based products included in the following sections."

Refer to R1C8 and R1C12 for the discussion on the seasonality of the proxies.

**R2C4**

Another point which I feel should be elaborated on is how independent are the separate sources of information? I assume there are many proxies which are used by the drought atlases and the two reanalysis products. Also the strengths and weaknesses of the different sources of information could be expanded upon so the reader gets a better idea of which is more reliable and where.

Refer to comment R1C3 for the discussion on the independence of the proxies of Fig. 1.

Refer to comment R1C10 for the proxy data assimilated into PHYDA/LMR. As discussed in that comment, the LMR and the PHYDA are based on the proxies from Steiger et al. (2018a). This has been clarified in the new Fig. 2 ("Coverage of the MCA and LIA by the proxies from Steiger et al. (2018a), considered for the generation of PHYDA and LMR.").

The new Fig. 2 is also used to discuss the reliability of the reconstruction-based products depending on the region (P4 L99: "Even if the PHYDA and LMR provide data for the whole millennium, some of the proxies used for the assimilation do not extend so far in the past. Figure 2 shows the distribution of proxies from Steiger et al. (2018a), including those that cover the MCA and LIA. There is a large amount of proxies covering these periods in regions like North America and Europe, so for these regions the products would be expected to follow the behavior of the proxies. However, for other regions like central Africa and eastern South America, the lack of proxies could increase the contribution of the model used for the assimilation, and thus the uncertainty of the products.").

As discussed in R1C13, the PHYDA and LMR use multiple types of proxies and the DAs are only based on tree rings. Even with that, most of the records used in the DAs are also included in the PHYDA/LMR, so the selection of proxies is not strongly different for both products.

**R2C5**

I think that more analysis could be conducted to truly separate external from internal variability. For example in the abstract you claim that the contribution of internal variability is particularly important in the Indo-Pacific basin. I do not see at the moment firm evidence for this in the document as it stands. I assume this is based on the model discrepancy in figure 6 as well as weak

ensemble mean signals. But its quite hard to rule out model differences in CMIP5 and CMIP6. Maybe a figure could be added explicitly looking at signal vs noise in the CESM-LME, to highlight where the external forcing is particularly important – at least in this model.

To deeper analyse the role of internal variability, the same analysis as for the CMIP5 and CMIP6 simulations have been done for the CESM-LME (Fig. 5c). In this figure, it can be seen that the number of simulations from the CESM-LME showing the same sign of the PDSI differences is small for areas of the Indo-Pacific basin. Since the simulations of CESM-LME share the forcing conditions and differ only in the internal variability, this implies that the internal variability plays a major role in these areas.

The text has been modified to describe this (P16 L229: "There is also no clear agreement for these areas among the individual simulations comprising the CMIP5 and CMIP6 ensembles (Fig. 5b), suggesting that the hydroclimate of these areas is strongly impacted by simulation-dependent processes, likely those associated with internal variability. The link to the internal variability is clear when analysing the simulations of the CESM-LME (Fig. 5c), which share the forcing conditions and differ only in the internal variability. These results are in line with the results obtained with the CESM-LME by Roldán-Gómez et al. (2022), showing a relevant impact of internal variability on the position and extension of the Indian and western Pacific ITCZ.")

Other specific points:
**R2C6**
Table 3 – HadCM -> HadCM3. Possibly best citation is Schurer et al 2014. https://doi.org/10.1038/ngeo2040

Reference to HadCM has been replaced by HadCM3. The citation has been also replaced by that of Schurer et al. (2014).

Refer to R1C11 for more details on the HadCM simulations.

**R2C7**
P11. You cite Luterbacher et al 2012 for fig 1. What is the relationship between the two?

There is no relationship between the Fig. 1 and the citation to Luterbacher et al (2012). The citation was added because the same behavior (drier conditions during the LIA in central and western Europe) is found both in the Fig. 1 and in the reference, but there is no relationship between them. To avoid misunderstandings on this, the reference to Luterbacher et al (2012) has been removed from this sentence and kept only in the first paragraph of the introduction, based only on the literature.

**R2C8**
Fig 4 – It seems quite confusing that you have placed on a model figure what I assume are proxy estimate of shifts in the ITCZ . Wouldn't it be better to put the multi model mean instead? Would that be possible? Feel free to ignore this comment if it is too much work…

The position of the ITCZ from models could be more representative for Fig. 4, but it would make the comparison with Fig. 3 more difficult. To allow for an easier comparison of the figures obtained with proxy data and those obtained with model simulations, we prefer to use the same position of the ITCZ in all of them (the one from Newton et al., 2006).

**R2C9**
Fig 6 – the scale is a bit confusing – if there are 3 positive and 3 negative models for example what would the color be? Would it be 3 or zero. Could you clarify this and perhaps consider whether a percentage agreement would be clearer. Also could you give an idea as to the number of products and models in each panel, in particular for panel b.

If there are 3 positive and 3 negative models a value of 3 was assigned. As this can be confusing, the scale of the figures has been modified, to include:

- Only those grid points with 4 or more products showing the same sign of PDSI difference for the Fig. 5a.
- Only those grid points with 8 or more simulations showing the same sign of PDSI difference for the Fig. 5b.

The total number of products and simulations has been added to the figure caption ("Number of products, from the 6 analysed products (DAs, PHYDA, LMR, and the ensemble averages of CESM-LME, CMIP5 and CMIP6)", "Number of simulations, from the 14 simulations in the ensembles of CMIP5 and CMIP6")

**R2C10**
Section 3.4. – could you give an indication of how much of this correlation is coming from proxy information and how much from the model fields which underline the reanalysis.

As discussed in R1C25, to analyse whether the teleconnections in reconstruction-based products come from proxies or from the model used in the assimilation process, the same correlation map have been added for the CESM-LME (Fig. 8c). The correlations obtained with the PHYDA strongly differ from those obtained with the CESM-LME, the model used in the data assimilation, showing that the contribution of the proxies to the teleconnections is more important than that of the model.

The text has been modified to comment on this (P21 L293: "The spatial correlations found in the PHYDA and LMR are mainly driven by the model used for the data assimilation. However, if the correlations are computed for the CESM-LME (Fig. 8c) without any data assimilation, only positive correlations associated to the impact of temperature in the PDSI are obtained").

---

## Author Response (AR2)

**Responses to reviewer's comments for "Model and proxy evidence for coordinated changes in the hydroclimate of distant regions over the Last Millennium"**

We are grateful to the reviewers for their comments and suggestions, all of which have been helpful for improving the manuscript. We respond to each of the comments in our thorough replies below, providing in gray the comments from each review and in black our responses.

**Reviewer 1:**

The authors resolved all points raised by the reviewer concerning the first submission. The manuscript does not have the traditional Results and Discussion sections. Instead, the Results section has various subsections in which the results and discussion are clearly separated. For this paper, I find this structure useful and clear. Hence, I suggest publishing the manuscript after the following two minor details have been addressed:

**R1C1**
1. Arrows in Fig. 1 are still difficult to understand. Do they just indicate the direction of change? But does their varying length have a meaning? Please add at least some explanation to the figure caption.

Arrows in Fig. 1 only indicate the direction of the change. The arrow length is arbitrarily selected to improve the visualisation of the figure.

This has been clarified in the figure caption: "The current position of the ITCZ in July and January, the ITCZ changes in the transition from MCA to LIA in the Indian Monsoon region (C1.ITCZ), Eastern Pacific (C2.ITCZ), East Africa (C3.ITCZ) and Western Pacific (C4.ITCZ), the boundary between low and high pressures in the pattern of NAM and SAM, and the changes in this boundary in the transition from MCA to LIA (C.NAM and C.SAM) are shown within the map, according to the references included in Table 1. Arrows only indicate the direction of the changes, and not their magnitude."

**R1C2**
2. Sec. 4.4 page 20: when you discuss the distant teleconnection, you should be a bit cautious and discuss the limitations of LMR and PHYDA. Both are based on fields from climate models and use the climatological covariance of the underlying models. Although using some covariance localization, they allow for distant observations to influence distant regions, which are teleconnected through the model covariance. This is why I would expect them to be in line with simulations. Only the drought atlases would be really independent reconstructions.

Indeed, the results of Fig. 8a and Fig. 8b are significantly impacted by the climatological covariance of the models. The last paragraph of section 4.4 has been reworded, to emphasize this limitation of the analyses based on PHYDA and LMR:

"The correlations for the PHYDA and LMR shown in Fig. 8a and Fig. 8b are impacted by the climatological covariance of the model used for the data assimilation. However, if the correlations are computed for the CESM-LME (Fig. 8c) without any data assimilation, only positive correlations associated to the impact of temperature in the PDSI are obtained, showing that a certain impact of the proxies may also be present in the results for the PHYDA and LMR. The fact that the PHYDA and LMR respectively emphasize the variability in the Pacific and Atlantic basin also shows that the regional variability given by the proxies is consistent with the spatial patterns from the models."

**Reviewer 2:**

I am happy that the reviewers have addressed my points, so my recommendation is to accept the paper for publication.

**R2C1**

My only comment is that the green colors on figure 5 (and several of the figures in the appendix) all look quite similar to me - so I wonder if it would be possible to bring out the differences in the agreement more by changing the color scale slightly.

The color scale of Fig. 3 to Fig. 5 was selected to use the colors of Fig. 1 as maximum value of the scale. This leaves indeed little room for the intermediate colors. However, this approach was preferred to a larger color scale decoupled from Fig. 1, since the conclusions are mostly extracted from the sign of the differences and the comparison of this sign with that from reconstructions of Fig. 1, rather than from the exact value of the PDSI differences.